

# Constructing stage-structured matrix population models from life tables: comparison of methods

Masami Fujiwara and  Jasmin Diaz-Lopez

Department of Wildlife and Fisheries Sciences, Texas A&M University, College Station, TX, United States of America

## ABSTRACT

A matrix population model is a convenient tool for summarizing *per capita* survival and reproduction rates (collectively vital rates) of a population and can be used for calculating an asymptotic finite population growth rate ($\lambda$) and generation time. These two pieces of information can be used for determining the status of a threatened species. The use of stage-structured population models has increased in recent years, and the vital rates in such models are often estimated using a life table analysis. However, potential bias introduced when converting age-structured vital rates estimated from a life table into parameters for a stage-structured population model has not been assessed comprehensively. The objective of this study was to investigate the performance of methods for such conversions using simulated life histories of organisms. The underlying models incorporate various types of life history and true population growth rates of varying levels. The performance was measured by comparing differences in $\lambda$ and the generation time calculated using the Euler-Lotka equation, age-structured population matrices, and several stage-structured population matrices that were obtained by applying different conversion methods. The results show that the discretization of age introduces only small bias in $\lambda$ or generation time. Similarly, assuming a fixed age of maturation at the mean age of maturation does not introduce much bias. However, aggregating age-specific survival rates into a stage-specific survival rate and estimating a stage-transition rate can introduce substantial bias depending on the organism's life history type and the true values of $\lambda$. In order to aggregate survival rates, the use of the weighted arithmetic mean was the most robust method for estimating $\lambda$. Here, the weights are given by survivorship curve after discounting with $\lambda$. To estimate a stage-transition rate, matching the proportion of individuals transitioning, with $\lambda$ used for discounting the rate, was the best approach. However, stage-structured models performed poorly in estimating generation time, regardless of the methods used for constructing the models. Based on the results, we recommend using an age-structured matrix population model or the Euler-Lotka equation for calculating $\lambda$ and generation time when life table data are available. Then, these age-structured vital rates can be converted into a stage-structured model for further analyses.

Corresponding author
Masami Fujiwara, fujiwara@tamu.edu

# INTRODUCTION

A matrix population model is a convenient tool for summarizing *per capita* survival and reproduction rates (collectively vital rates) of a population, and is used for calculating an asymptotic finite population growth rate (commonly denoted by $\lambda$) and generation time. These two pieces of information can be used for determining the status of a threatened species (*IUCN, 2012*). Matrix population models are broadly categorized into age-structured (Leslie matrix; *Leslie, 1945*) and stage-structured (Lefkovitch matrix; *Lefkovitch, 1965*) models. Age-structured matrix models group individuals based on age whereas stage-structured matrix models group individuals based on other properties such as developmental stage and size. Stage-structured matrix models are often favored when a property of individuals besides age is a better indicator of survival and reproduction (e.g., *Caswell, 2001*; *Cochran & Ellner, 1992*). Although both types of models are common, the use of stage-structured population models has increased in recent years. Moreover, in many studies, vital rates in stage-structured population models may be estimated from age-structured vital rates through the grouping of age-classes together to form a stage (e.g., *Brault & Caswell, 1993*; *Caswell et al., 1998*; *Crouse, Crowder & Caswell, 1987*; *Crowder et al., 1994*). Typically a life table analysis is one of the main methods for obtaining age-specific vital rates.

A life table is a list of age-specific population density and age-specific fecundity. From life table data, survivorship (the proportion of age 0 that is alive at age $x$) is estimated, and then age-specific survival rates (the proportion of individuals that survive from age $x$ to $x + 1$) are estimated. Age-specific survival rates along with age-specific fecundity rates can be entered into an age-structured matrix population model almost directly (see the Age-Structured Matrix Population Models section), and $\lambda$ and generation time can be calculated from the matrix. Matrix population models are also used for sensitivity and elasticity analyses (*Caswell, 1978*; *De Kroon, Groenendael & Ehrlen, 2000*). The inclusion of a large number of age-classes for long-lived organisms can make the interpretation of the sensitivity and elasticity analyses complicated because individuals in multiple age classes are often practically identical but separated in an age-structured model. Consequently, when long-lived organisms are studied, it is common to convert age-specific vital rates into stage-specific vital rates, and to use stage-structured population matrices for calculating $\lambda$ and generation time.

In order to convert age-specific into stage-specific vital rates, the former vital rates need to be aggregated for the various stages. Furthermore, a transition rate from one stage to another needs to be estimated. Several approaches, or *conversion methods*, to aggregate survival rates for calculating a transition rate exist. However, the performance of the conversion methods has not been investigated comprehensively. Intuitively, the performance should depend on how survivorship and reproductive schedule change with age (life history) and whether a population is growing or declining in its abundance. Therefore, it is critically important to investigate the performance of the conversion methods for different life history and population growth scenarios.

The objective of this study is to investigate the performance of the conversion methods to estimate vital rates for stage-structured matrix population models from life tables. The performance is measured by comparing $\lambda$'s and generation time calculated with the Euler-Lotka equation (see *Kot, 2001*), age-structured population matrices, and several stage-structured population matrices obtained using different conversion methods. The asymptotic finite population growth rate ($\lambda$) and the generation time calculated with the Euler-Lotka equation are considered the true values. Any discrepancies between the results obtained from the Euler-Lotka equation and those from an age-structured population matrix are considered to be due to bias introduced by discretizing age. In contrast, discrepancies between results from an age structured population matrix and those from stage-structured population matrices are considered to be due to bias introduced by conversion methods.

The comparisons were carried out with the life table data of organisms with different life history types. A wide range of life history types was incorporated into the analysis by artificially creating them using a competing risk model (*Siler, 1979*) and three types of fecundity functions. However, the analysis focuses on those with a prolonged duration in a stage of interest. When the duration is short, stage structure is often embedded within age structure, and it is not necessary to convert age-structured vital rates into stage-structured rates. The comparisons were also repeated with life table data from populations that are declining ($\lambda < 1$), maintaining the same population density ($\lambda = 1$), or increasing ($\lambda > 1$) in population density in order to determine whether a stage-structured population matrix can be used for determining a population growth rate correctly under different population growth conditions.

This paper is structured as follows. First, the life history types of organisms considered and the conversion methods to be compared are described. Then, methods for calculating $\lambda$'s and generation time with the Euler-Lotka equation and matrix population models are described followed by the description of the procedures specific to this study. Finally, results are presented and discussed.

## MODELS AND METHODS

### Life history types

The types of life histories considered are broadly categorized into two groups: one with a short juvenile stage (early maturation) and the other with a prolonged juvenile stage (delayed maturation). The early maturation type was used for comparing methods for converting age-specific survival rates and fecundity rates into a stage-specific adult survival rate and a stage-specific fertility rate. The delayed maturation type was used for comparing methods for converting age-specific survival rates into a stage-specific juvenile survival rate and a stage transition rate. Throughout this study, the unit for time and age was set to a year for convenience, but it can be scaled differently if both are changed in the same way. Age is denoted by $x$, and organisms experience five major life events: birth at age 0, beginning of the juvenile stage at age $x_0$, age at which the first individual mature $x_1$, the
**Table 1  Parameters for twelve different life history types.**

| Model | $\alpha_1$ | $\alpha_2$ | $\beta_2$ | $L_\infty$ | $\kappa$ | $R^{-1}_{\lambda=0.9}$ | $R^{-1}_{\lambda=1.0}$ | $R^{-1}_{\lambda=1.1}$ | $x_0$ | $x_1$ | $x_2$ | $x_3$ |
|---|---|---|---|---|---|---|---|---|---|---|---|---|
| **Early maturation type** | | | | | | | | | | | | |
| Dimension | $\tau^{-1}$ | $\tau^{-1}$ | $\tau^{-1}$ | $l$ | $\tau^{-1}$ | $\mu^{-1}\tau l^3$ | $\mu^{-1}\tau$ | $\mu^{-1}\tau$ | $\tau$ | $\tau$ | $\tau$ | $\tau$ |
| CH-IF | .1535 | 0 | – | 10 | .10 | 6,832 | 948 | 280 | 1.5 | 1.5 | 1.5 | 40.5 |
| CH-CF | .1535 | 0 | – | 10 | – | 16,371 | 5,163 | 2,768 | 1.5 | 1.5 | 1.5 | 40.5 |
| CH-DF | .1535 | 0 | – | 10 | .10 | 6,262 | 3,134 | 1,975 | 1.5 | 1.5 | 1.5 | 40.5 |
| IH-IF | 0 | .002 | .205 | 10 | .10 | 39,459 | 5,985 | 1,435 | 1.5 | 1.5 | 1.5 | 40.5 |
| IH-CF | 0 | .002 | .205 | 10 | – | 80,276 | 18,485 | 7,244 | 1.5 | 1.5 | 1.5 | 40.5 |
| IH-DF | 0 | .002 | .205 | 10 | .10 | 22,872 | 8,100 | 4,176 | 1.5 | 1.5 | 1.5 | 40.5 |

| Model | $\alpha_1$ | $\alpha_3$ | $\beta_3$ | $m_1$ | $m_2$ | $a$ | $R^{-1}_{\lambda=0.9}$ | $R^{-1}_{\lambda=1.0}$ | $R^{-1}_{\lambda=1.1}$ | $x_0$ | $x_1$ | $x_2$ | $x_3$ |
|---|---|---|---|---|---|---|---|---|---|---|---|---|---|
| **Delayed maturation type** | | | | | | | | | | | | | |
| Dimension | $\tau^{-1}$ | $\tau^{-1}$ | $\tau^{-1}$ | $g\tau^{-1}$ | $\tau^{-1}$ | $\tau^{-1}$ | $\mu^{-1}\tau l^3$ | $\mu^{-1}\tau$ | $\mu^{-1}\tau$ | $\tau$ | $\tau$ | $\tau$ | $\tau$ |
| DH-FM | .1000 | .400 | .300 | – | – | .20 | 3.000 | 0.497 | 0.124 | 1.5 | 10.5 | 10.5 | 40.5 |
| CH-FM | .2193 | 0 | – | – | – | .20 | 3.005 | 0.498 | 0.124 | 1.5 | 10.5 | 10.5 | 40.5 |
| DH-CM | .1000 | .400 | .300 | .50 | 0 | .20 | 2.958 | 0.502 | 0.131 | 1.5 | 8.5 | 10.5 | 40.5 |
| CH-CM | .2193 | 0 | – | .50 | 0 | .20 | 3.082 | 0.538 | 0.143 | 1.5 | 8.5 | 10.5 | 40.5 |
| DH-IM | .1000 | .400 | .300 | .10 | 1.01 | .20 | 2.942 | 0.491 | 0.124 | 1.5 | 8.5 | 10.5 | 40.5 |
| CH-IM | .2193 | 0 | – | .10 | 1.01 | .20 | 3.030 | 0.510 | 0.130 | 1.5 | 8.5 | 10.5 | 40.5 |

**Notes.**

CH, Constant hazard (risk) model (Eq. (1)); IH, Increasing hazard (risk) model (Eq. (2)); DH, Decreasing hazard (risk) model (Eq. (6)); IF, Increasing fecundity model (Eq. (3)); CF, Constant fecundity model (Eq. (4)); DF, Declining fecundity model (Eq. (5)); FM, Fixed age of maturation at age $x_2$; CM, Constant rate of maturation; IM, Exponentially increasing rate of maturation; $\tau$, age; $l$, length; $\mu$, number of births per adult; $g$, number of individuals reaching maturity per juvenile.

mean age of maturation $x_2$, and the last age of reproduction $x_3$. Hereafter, the subscripts for age $x$ denotes life events. The parameters for all models to be described are listed in Table 1.

### Early maturation types

For the early maturation types, two types of survivorship and three types of fecundity schedules were incorporated. For the first type of survivorship, individuals experience a constant risk of mortality. This leads to an exponentially declining survivorship curve ($l(x)$) with age:

$$l(x) = \mathrm{e}^{-\alpha_1 x}, \tag{1}$$

where $\alpha_1$ is the constant risk (i.e., hazard rate). For the second type of survivorship, individuals experience an exponentially increasing risk of mortality with age:

$$l(x) = \mathrm{e}^{\frac{\alpha_2}{\beta_2}\left(1 - \mathrm{e}^{\beta_2 x}\right)}, \tag{2}$$

where $\alpha_2$ and $\beta_2$ are the parameters for the increasing risk. Under this model, the hazard function is given by $h(x) = \alpha_2 e^{\beta_2 x}$ (*Siler, 1979*). This type of risk may be due to aging. These two types of survivorship curves are shown in Fig. 1A.

Figure 1B shows the three types of fecundity schedules. The first type assumes the fecundity $b(x)$ is proportional to the cube of the body length of the individuals, which in
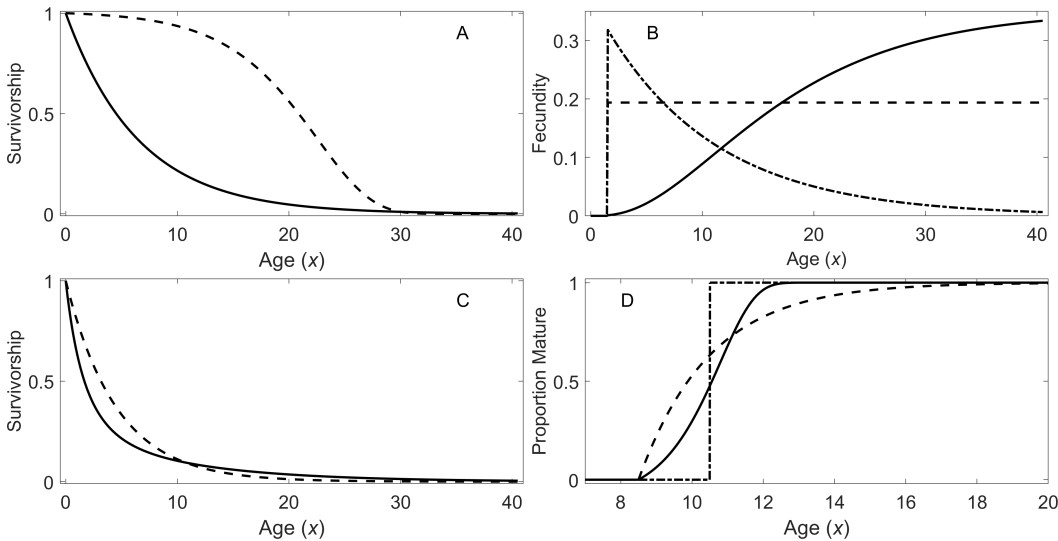

**Figure 1 Survivorship, fecundity, and maturation schedule.** (A) Survivorship of individuals under a constant risk (solid line, Eq. (1)) and an exponentially increasing risk (dotted line, Eq. (2)). (B) Fecundity as a function of age: increasing fecundity (solid line; Eq. (3)), constant fecundity (dotted line; Eq. (4)), and exponentially declining fecundity (dash-dot; Eq. (5)). (C) Survivorship of individuals under a declining risk (solid line; Eq. (6)) and a constant risk (dashed line; Eq. (1)). (D) Proportion of individuals mature with the increasing rate of maturation (solid line), the fixed age of maturation (dotted line), and the constant rate of maturation (dash dot).

turn increases according to von Bertalanffy growth equation:

$$b(x) = R_1 \left[ L_\infty \left( 1 - e^{-\kappa x} \right) \right]^3, \tag{3}$$

where $L_\infty$ and $\kappa$ are the parameters in the growth model (*Von Bertalanffy, 1968*). This assumes reproduction is approximately proportional to the mass of individuals. The second type assumes a constant fecundity with age

$$b(x) = R_2. \tag{4}$$

Finally, the third type assumes exponentially declining fecundity:

$$b(x) = R_3 e^{-\gamma x}, \tag{5}$$

where $\gamma$ is the parameter determining how fast fecundity declines with age. Parameter $R_m$ (where $m = 1, 2,$ or $3$) are constants with different units depending on the type of fecundity, and these constants were used for adjusting $\lambda$'s in this study (see 'Analytical Procedure'). For all of the early maturation types in this study, individuals are assumed to mature at age 1.5 (i.e., $x_1 = x_2 = 1.5$).

### Delayed maturation types

For the delayed maturation types, two kinds of survivorship curves are incorporated into a juvenile stage. One incorporates a constant risk of mortality (Eq. (1)), which results in exponentially declining survivorship (dashed curve in Fig. 1C). The other assumes an

exponentially declining risk of mortality in addition to a constant risk:

$$l(x) = e^{-\alpha_1 x} e^{\left[ -\frac{\alpha_3}{\beta_3}\left(1 - e^{-\beta_3 x}\right) \right]}, \tag{6}$$

where $\alpha_3$ and $\beta_3$ are the parameters for the declining risk. In this model, the exponentially declining hazard is given by $h(x) = \alpha_3 e^{-\beta_3 x}$ (*Siler, 1979*). This causes initial fast descent of survivorship (solid curve in Fig. 1C). This type of survivorship can be observed when organisms grow out of a mortality risk such as predation as they age. Once individuals mature, they reach a constant risk of mortality (Eq. (1)) and constant fecundity (Eq. (4)) under the delayed maturation types.

In addition, three types of maturation schedule are incorporated. The first type assumes a fixed age of maturation at age 10.5 years ($x_1 = x_2 = 10.5$). The second type assumes a constant rate of maturation, and the third type assumes an exponentially increasing rate of maturation with age. In the latter two cases, maturation begins at age 8.5 years ($x_1 = 8.5$), and the parameters are chosen so that the mean age of maturity is approximately 10.5 years ($x_2 = 10.5$); however, the mean age varies slightly depending on the level of mortality rate during the maturation period. Under the latter two types of maturation schedules, the densities of individuals in immature (all stages prior to maturation) and adult stages are described conveniently with a system of ordinary differential equations (ODEs):

$$\begin{aligned} \frac{dn_1(x)}{dx} &= -m(x)n_1(x) - h(x)n_1(x) \\ \frac{dn_2(x)}{dx} &= m(x)n_1(x) - an_2(x) \end{aligned}, \tag{7}$$

where $n_1(x)$ and $n_2(x)$ are the densities of immature individuals and adults at age $x$, respectively, $m(x)$ is a per-capita instantaneous maturation rate at age $x$, and $h(x)$ is the age-dependent hazard function of mortality. For the second type of maturation schedule, $m(x)$ is set at $m_1 = 0.5$ for $x \geq x_1$ and 0 otherwise. For the third type, $m(x) = m_1 e^{m_2(x - x_1)}$ for $x \geq x_1$, and $m(x) = 0$ otherwise. The proportion of individuals that are mature at a given age for the three types of maturation schedule is shown in Fig. 1D. When the system of ODEs is solved with the initial condition $[n_1(0) \quad n_2(0)] = [1 \quad 0]$ using a numerical ODE solver, the survivorship is given by $l(x) = n_1(x) + n_2(x)$. Note Eq. (7) does not include reproduction, and the initial condition is 1 in the immature stage; therefore, $l(x) \leq 1$ for all $x \geq 0$.

## Conversion methods

In order to convert age-structured vital rates into stage-structured vital rates, three types of vital rates need to be calculated: a stage-specific survival rate for stage $i$ ($S_i$), a transition rate from stage $i$ to stage $j$ conditional on their survival ($P_{j,i}$), and a fertility rate for stage $i$ ($F_i$). The methods for converting these vital rates are described in the following sections.

### Stage-specific survival rate

All the methods for calculating a stage-specific survival rate from age-specific survival rates investigated in this study assume that the beginning and ending ages of a stage are given by the mean ages of transition into and from the stage, respectively, and that these

ages are known. The former assumption can potentially produce bias when calculating $\lambda$ and generation time because stage transitions often do not occur at a fixed age (i.e., individuals gradually transition from one stage to another over a range of ages). Therefore, the performance of conversion methods was also investigated when the age of stage-transitions was not fixed (Eq. (7)). It is also possible that stage-transitions are completely age-independent (e.g., as in some plant species). This situation is beyond the scope of the current analysis, but it will be briefly discussed in the 'Discussion' section.

An age-specific survival rate from age $x$ to $x+1$ is given by

$$s_x = \frac{l(x+1)}{l(x)}. \tag{8}$$

Hereafter, lower case $s$ is used for denoting an age-specific survival rate whereas the upper case $S$ is used for denoting a stage-specific survival rate. Suppose the stage begins at age $x^{(i)}$ and ends at $x^{(j)} - 1$ (i.e., age $x^{(j)}$ is the next stage), then the survival rate of individuals in the stage is given by the mean survival rate over age classes within the stage. Note that an index in the superscript within parentheses denote a stage. Here, nine different ways of calculating the mean are compared. First, the mean is given by geometric mean, arithmetic mean, or harmonic mean (Table 2). These are three common methods for obtaining a stage-specific survival rate. However, the number of individuals in age classes are different, and it is more appropriate to put weight based on the proportion of individuals in a given age class, which is given by a survivorship curve. This leads to a weighted geometric mean, weighted arithmetic mean, and weighted harmonic mean (Table 2). Although weighting with survivorship is fine when a population is neither growing nor declining (i.e., $\lambda = 1$), when a population is growing (or declining), there are more (or less) individuals in younger age classes than predicted by a survivorship curve. Therefore, it is necessary to discount the weight using $\lambda$. This leads to weighted means where the weight is given by both survivorship and $\lambda$ (Table 2). Hereafter, the latter weight is termed "a discounted weight."

### Conditional transition rate

Transition rate from stage $i$ to stage $j$ conditional on their survival ($p_{j,i}$) is calculated in three ways (Table 3). The first method (T1) matches the expected duration in the stage assuming exponentially declining time to transition into the following stage. The second method (T2) matches the expected proportion of individuals making the transition into the following stage assuming the survivorship curve gives the distribution of individuals among age-classes. The third method (T3) is the same as the second method except that the distribution is discounted by a population growth rate. These methods are described in more detail in *Caswell (2001)*.

### Fertility rate

Fertility rates in matrix population models are different from fecundity in a life table. A fertility rate gives the *per capita* rate of contribution from one stage to the next over one year so that adults will have to survive to reproduce, and/or offspring must survive to appear in the next stage. On the other hand, fecundity is the number of offspring produced. The latter does not include any survival of adult or offspring. Consequently, there are

**Table 2  Nine ways of converting age-specific survival rates into a stage-specific survival rate.** Note lower-case $s$ is used for an age-specific survival rate and upper-case $S$ is used for a stage-specific survival rate. $x_i$ is the first age class in the stage $i$, and $x_j$ is the first age class in stage $j = i + 1$.

| Type of mean | Formula |
|---|---|
| Geometric | $S_i = \left( \prod_{x=x_i}^{x_j - 1} s_x \right)^{\frac{1}{(x_j - x_i)}}$ |
| Arithmetic | $S_i = \frac{1}{x_j - x_i} \sum_{x=x_i}^{x_j - 1} s_x$ |
| Harmonic | $S_i = \left( \frac{\sum_{x=x_i}^{x_j - 1} s_x^{-1}}{x_j - x_i} \right)^{-1}$ |
| Weighted geometric | $S_i = \left( \prod_{x=x_i}^{x_j - 1} s_x^{\omega_{1,x}} \right)^{\frac{1}{\sum_{x=x_i}^{x_j - 1} \omega_{1,x}}}$ where $\omega_{1,x} = l(x)$ |
| Weighted arithmetic | $S_i = \frac{\sum_{x=x_i}^{x_j - 1} \omega_{1,x} s_x}{\sum_{x=x_i}^{x_j - 1} \omega_{1,x}}$ where $w_{1,x} = l(x)$ |
| Weighted harmonic | $S_i = \left( \frac{\sum_{x=x_i}^{x_j - 1} \omega_{1,x} s_x^{-1}}{\sum_{x=x_i}^{x_j - 1} \omega_{1,x}} \right)^{-1}$ where $\omega_{1,x} = l(x)$ |
| Discounted-weight geometric | $S_i = \left( \prod_{x=x_i}^{x_j - 1} s_x^{\omega_{2,x}} \right)^{\frac{1}{\sum_{x=x_i}^{x_j - 1} \omega_{2,x}}}$ where $\omega_{2,x} = \frac{1}{\lambda^{x - x_i}} l(x)$ |
| Discounted-weight arithmetic | $S_i = \frac{\sum_{x=x_i}^{x_j - 1} \omega_{2,x} s_x}{\sum_{x=x_i}^{x_j - 1} \omega_{2,x}}$ where $\omega_{2,x} = \frac{1}{\lambda^{x - x_i}} l(x)$ |
| Discounted-weight harmonic | $S_i = \left( \frac{\sum_{x=x_i}^{x_j - 1} \omega_{2,x} s_x^{-1}}{\sum_{x=x_i}^{x_j - 1} \omega_{2,x}} \right)^{-1}$ where $\omega_{2,x} = \frac{1}{\lambda^{x - x_i}} l(x)$ |

**Table 3  Three methods for the calculation of the conditional stage-transition rate $P_{j,i}$.** $x_i$ is the first age class in the stage $i$, and $x_j$ is the first age class in stage $j = i + 1$.

| MODEL | Type of transition rate | Formula |
|---|---|---|
| T1 | Matching duration | $P_{j,i} = 1 - \frac{1}{x_j - x_i}$ |
| T2 | Matching proportion transitioning | $P_{j,i} = \frac{l(x_j - 1)}{\sum_{x=x_i}^{x_j - 1} l(x)}$ |
| T3 | Matching proportion transitioning with discount | $P_{j,i} = \frac{\lambda^{-(x_j - x_i - 1)} l(x_j - 1)}{\sum_{x=x_i}^{x_j - 1} \lambda^{-(x - x_i)} l(x)}$ |

two steps in converting life table data into a stage-structured fertility rate: calculating age-specific fertility rates and converting them into a stage-specific fertility rate. There are many approaches for the first step depending on the reproductive schedules of organisms within a year (see *Caswell, 2001*) or among years (e.g., *Crowder et al., 1994*). Evaluating

them is beyond the scope of this study. Here, reproduction was assumed to occur at any time of year (i.e., a birth flow model), and an age-specific fertility rate ($f_x$) was estimated as follows:

$$f_x = l(0.5) \int_{z=x-0.5}^{x+0.5} b(z) \frac{l(z)}{l(x-0.5)} dz, \qquad (9)$$

where $b(z)$ is the instantaneous per-capita fecundity rate at age $x$. In the equation, $l(z)/l(x-0.5)$ gives the survival rate of adults from age $x-0.5$ to age $z$ (where $z > x-0.5$), and the integral calculates the total number of offspring produced between ages $x-0.5$ and $x+0.5$ per adult that was alive at age $x-0.5$. Then, all births are attributed to the half way point (i.e., age $x$) so that offspring will have to survive half a year on average to appear in the first stage; this survival rate is given by $l(0.5)$, which is the survival rate from age 0 to 0.5.

Once age-specific fertility rates are obtained, a stage-specific fertility rate $F_i$ is calculated in three different ways. First, an arithmetic mean is taken with equal weights on all age classes,

$$F_i = \frac{1}{x_3 - x_2} \sum_{x=x_2+0.5}^{x_3-0.5} f_x. \qquad (10)$$

The second approach is to take a weighted arithmetic mean,

$$F_i = \frac{1}{\sum_{x=x_2+0.5}^{x_3-0.5} \omega_{1,x}} \sum_{x=x_2+0.5}^{x_3-0.5} \omega_{1,x} f_x, \qquad (11)$$

where $\omega_{1,x} = l(x)$. The third method is to use a discounted weight by replacing $\omega_{1,x}$ in Eq. (11) with $\omega_{2,x} = \frac{1}{\lambda^{x-x_0}} l(x)$. In all calculations, the same weight (or no weight) is used for calculating a stage-specific survival rate and a stage-specific fertility rate.

## Asymptotic population growth rate and generation time

An asymptotic population growth rate (a finite *per capita* population growth rate $\lambda$) and generation time are calculated using the Euler-Lotka equation, age-structured (Leslie) matrices, and stage-structured (Lefkovitch) matrices. These models and the calculations of $\lambda$ and generation time are described in this section.

### *Euler-Lotka equation*

The Euler-Lotka equation (*Kot, 2001*) is given by

$$1 = \int_0^{x_3} \lambda^{-x} b(x) l(x) \, dx. \qquad (12)$$

The fecundity $b(x)$ and survivorship $l(x)$ are defined for a specific life history type as described in the previous section (Fig. 1). Provided $R_m$ is fixed, the only unknown is $\lambda$, which can be found numerically by searching the value that satisfies Eq. (12) using a root-finding algorithm.

Generation time in this study is defined as the mean age of parents where the mean is calculated over all offspring born at a given time. More specifically, generation time $G_1$ is

given by

$$G_1 = \int_0^{x_3} x\lambda^{-x} b(x) l(x)\, dx. \tag{13}$$

(*Keyfitz & Caswell, 2005*). In the case where individuals mature over a range of ages, some individuals do not reproduce in certain age ranges. Therefore, generation time $G_2$ is instead given by

$$G_2 = \int_0^{x_3} x\lambda^{-x} b(x) q(x) l(x)\, dx, \tag{14}$$

where $q(x)$ is the proportion of individuals that are mature at age $x$.

### Age-structured matrix population models (Leslie matrix)

An age-structured population matrix is given as

$$\mathbf{A} = \begin{bmatrix} 0 & f_2 & \cdots & f_{x_3-1.5} & f_{x_3-0.5} \\ s_{0.5} & 0 & \cdots & 0 & 0 \\ 0 & s_{1.5} & \cdots & 0 & 0 \\ \cdots & \cdots & \cdots & \cdots & \cdots \\ 0 & 0 & \cdots & s_{x_3-2} & 0 \end{bmatrix}. \tag{15}$$

In this study, the first age class is assumed to begin at age 0.5, and subsequent age class is incremented by one year. It should be noted that survival rate $s_x$ gives the proportion of individuals that survive from time $x$ to $x+1$ and fertility $f_x$ gives the mean number of offspring of age 0.5 produced by a parent alive at age $x - 0.5$ by reproducing between age $x - 0.5$ and $x + 0.5$ (see Eq. (9)). It is assumed that the reproduction begins at the mean age of maturation $x_2$.

In the case where individuals mature over a range of age, the population matrix is given instead as

$$\mathbf{A} = \begin{bmatrix} 0 & q_2 f_2 & \cdots & q_{x_3-1.5} f_{x_3-1.5} & q_{x_3-0.5} f_{x_3-0.5} \\ s_{0.5} & 0 & \cdots & 0 & 0 \\ 0 & s_{1.5} & \cdots & 0 & 0 \\ \cdots & \cdots & \cdots & \cdots & \cdots \\ 0 & 0 & \cdots & s_{x_3-2} & 0 \end{bmatrix}. \tag{16}$$

Once the population matrix is constructed, $\lambda$ is obtained by taking its dominant eigenvalue. Furthermore, generation time is given by

$$G_3 = \frac{\lambda \mathbf{v} \mathbf{w}}{\mathbf{v} \mathbf{F} \mathbf{w}} \tag{17}$$

where $\mathbf{v}$ and $\mathbf{w}$ are the left and right eigenvectors (row and column vectors), respectively, of the population matrix, and $\mathbf{F}$ is a modified population matrix only consisting of fertility terms (*Bienvenu & Legendre, 2015*).

### Stage-structured matrix population models (Lefkovitch matrix)

Stage-structured matrix population models are given in two different forms in this study. For early maturation types, a two-stage population matrix is used

$$\mathbf{A} = \begin{bmatrix} 0 & F_2 \\ S_1 & S_2 \end{bmatrix} \tag{18}$$

where the duration of stage 1 is from age $x_0 = 0.5$ to age $x_1 = x_2 = 1.5$ and the duration of stage 2 is from age 1.5 to age $x_3 = 40.5$.

For organisms with delayed maturation, three-stage matrix population models are used

$$\mathbf{A} = \begin{bmatrix} 0 & 0 & F_3 \\ S_1 & S_2(1-P_{3,2}) & 0 \\ 0 & S_2 P_{3,2} & S_3 \end{bmatrix}. \tag{19}$$

The duration in stages 1, 2, and 3 are assumed to be from age $x_0 = 0.5$ to 1.5, from age 1.5 to $x_1 = x_2 = 10.5$, and from age $x_2 = 10.5$ to $x_3 = 40.5$. Using the stage-structured matrix, $\lambda$ and generation time can be calculated in the same way as with the age-structured matrices.

## Analytical procedure

Thirty-six different scenarios of life history strategies and a population growth rate were investigated in this study. For the early maturation type, there were six different life history strategies (two types of survivorships and three types of fecundity schedules; Figs. 1A and 1B). With each of the six life history strategies, the true finite asymptotic population growth rate ($\lambda$) was adjusted to 0.900, 1.000, and 1.100 by adjusting coefficient $R_m$ in the fecundity function (Eqs. (3)–(5)). This adjustment was done with the Euler-Lotka equation by searching the value $R_m$ that gives the corresponding value of $\lambda$. Consequently, there were 18 early maturation type scenarios to investigate. For each scenario, one age-structured population model and nine two-stage population models were constructed; the nine models differ in the ways that adult survival rate and a fertility rate were calculated (see Table 2). Then, with each model, $\lambda$ and generation time were calculated.

Similarly, for the delayed maturation type, there were six different life history strategies (two types of survivorships and three types of maturation schedules; Figs. 1C and 1D). For each type, finite asymptotic population growth rate ($\lambda$) was also adjusted to 0.900, 1.000, and 1.100. Then, two age-structured population models (Eqs. (15) and (16)) and nine three-stage population models were constructed; these nine models differ in the ways that a juvenile survival rate was aggregated and a conditional transition rate was obtained (see Table 2). Then, with each model, $\lambda$ and generation time were calculated.

In order to obtain the true generation time, Eq. (13) was used when organisms had a fixed age of maturation, while Eq. (14) was used when organisms matured over a range of age. All of the calculations were done with *MathWorks (2012)*. For solving a system of ODE's, a built-in ODE solver "ode45" was used in most cases. When a maturation rate was exponentially increasing with age, there was a numerical problem with the function (equations became stiff). Therefore, "ode15s" was used, instead. For the methods that require $\lambda$ in aggregating a survival rate, a transition rate, and/or a fertility rate, the true $\lambda$ was used.

In actual analyses conducted with real data, estimating $\lambda$ is one of the main purposes of constructing a population matrix. In other words, it is not known *a priori*. To overcome this issue, *Caswell (2001)* describes an iterative method. In this method, an initial $\lambda$ is arbitrarily selected and used for obtaining factors needed to calculate the conditional transition rate. Then, $\lambda$ is estimated from the obtained population matrix. Subsequently, the new $\lambda$ is used
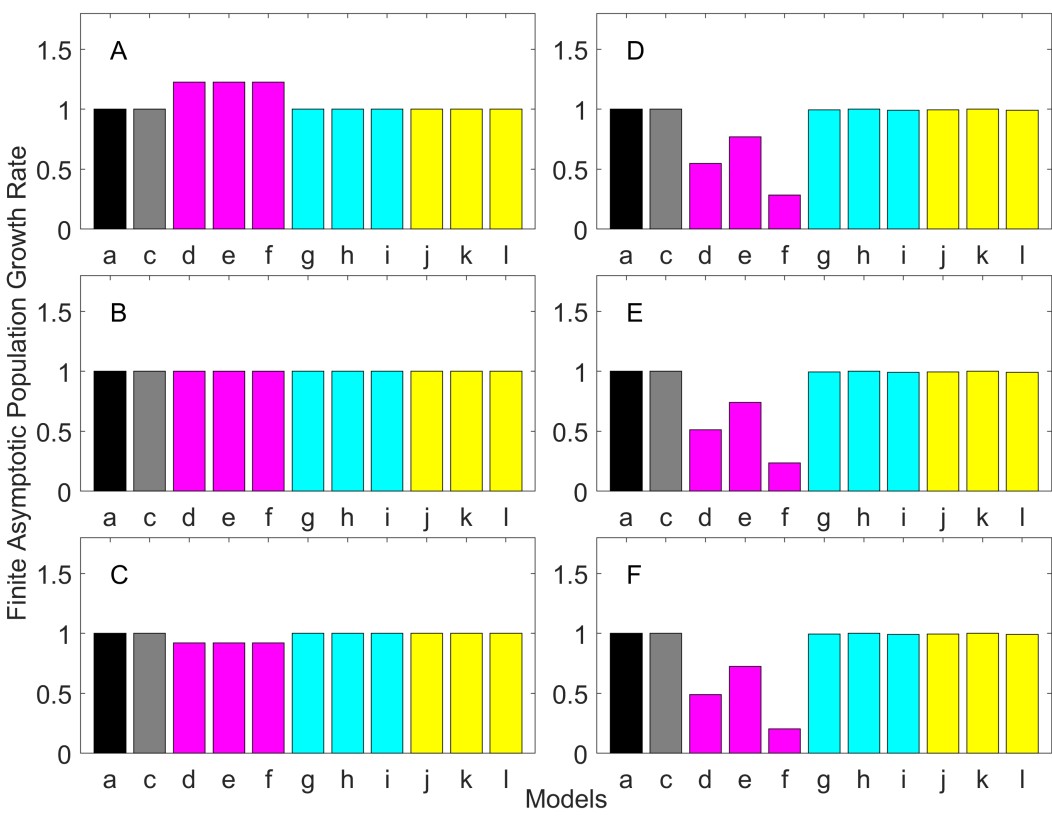

**Figure 2** **Finite asymptotic population growth rate λ when the true λ is 1.000 for early maturation types.** Each panel represents different life history type as was defined in Table 1. Each bar represents a different model for estimating λ. See Table 4 for model types. (Black, Euler-Lotka equation; Gray, age-structured model; Magenta, stage-structured with ordinary mean; Cyan, stage-structured with weighted mean; Yellow, stage-structured with weight mean with discount). (A) Constant Hazard and Increasing Fecundity, (B) Constant Hazard and Constant Fecundity, (C) Constant Hazard Decreasing Fecundity, (D) Increasing Hazard and Increasing Fecundity, (E) Increasing Hazard and Constant Fecundity, (F) Increasing Hazard and Decreasing Fecundity.

to obtain a new population matrix and λ is calculated again. This process is repeated until λ converges. For example, this method was used for developing a stage-structured population matrix of loggerhead sea turtles (*Crowder et al., 1994*). In the current study, this procedure is applied to delayed maturation life history strategy. Discounted-weight arithmetic mean was used for aggregating juvenile and adult survival rates and a transition rate was obtained by matching proportion transitioning also using λ as a discounting factor.

## RESULTS

### Finite asymptotic population growth rate

Figures 2–7 show estimated finite *per capita* population growth rates (λ). In each figure, each panel shows the estimated λ of the same life history type calculated using different methods. These methods are listed in Table 4. The first bar (black) is the true value, and the rest of the bars are obtained with age- or stage-structured population matrices.

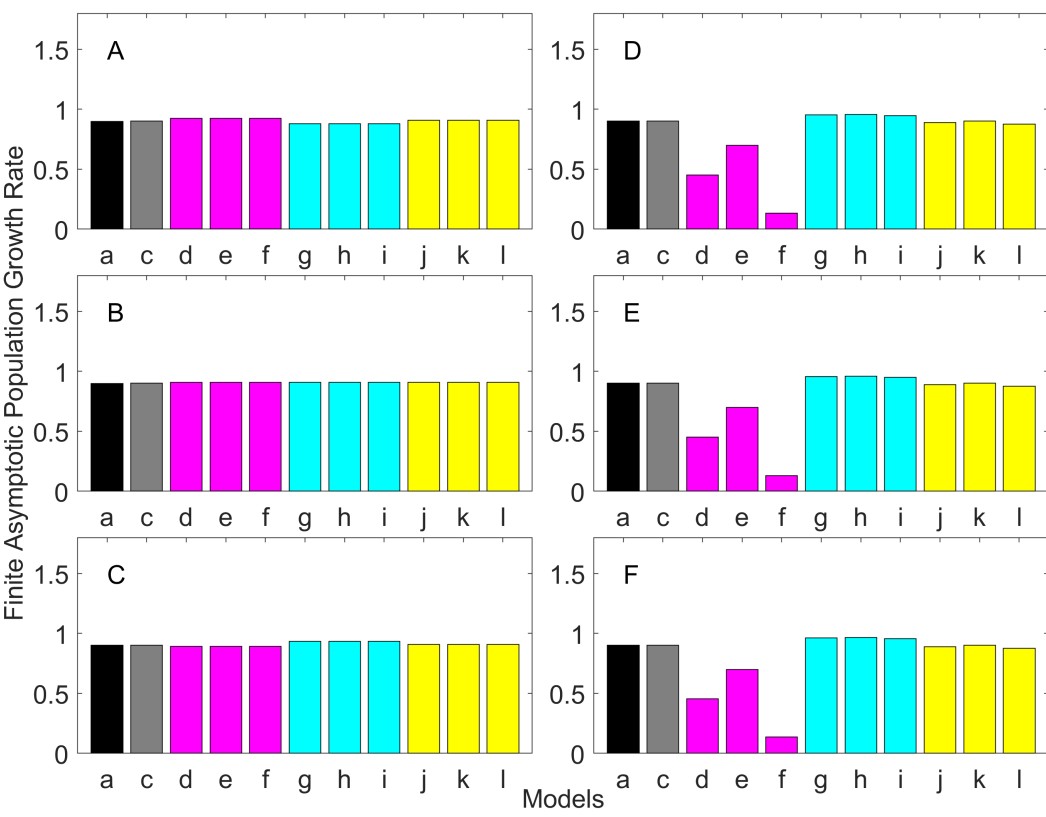

**Figure 3** **Finite asymptotic population growth rate λ when the true λ is 0.900 for early maturation types.** Each panel represents different life history type as was defined in Table 1. Each bar represents a different model for estimating λ. See Table 4 for model types. (Black, Euler-Lotka equation; Gray, age-structured model; Magenta, stage-structured with ordinary mean; Cyan, stage-structured with weighted mean; Yellow, stage-structured with weight mean with discount). (A) Constant Hazard and Increasing Fecundity, (B) Constant Hazard and Constant Fecundity, (C) Constant Hazard Decreasing Fecundity, (D) Increasing Hazard and Increasing Fecundity, (E) Increasing Hazard and Constant Fecundity, (F) Increasing Hazard and Decreasing Fecundity.

For the early maturation types (Figs. 2–4), one age-structured model and nine stage-structured models that differ in the methods for calculating an adult survival rate were used. The three figures are different in the true λ's. For all life history types and all true λ's (Figs. 2–4), the Euler-Lotka equation (Model a) and age-structured model (model c) show the similar values (black and gray bars in Figs. 2–4). This suggests there is no bias introduced by discretizing age.

When adults experience a constant risk of mortality (Eq. (1)) and constant fecundity (Eq. (4)), all conversion methods performed well (within 2% of the true value; see of Figs. 2B–4B). In all other cases, some of the methods performed poorly (see Figs. 2A, 2C–2F, 3A, 3C–3F and 4A, 4C–4F).

Comparing the use of the geometric mean, arithmetic mean, and harmonic mean of adult survival rate (Models d–f in Figs. 2A–2C, 3A–3C and 4A–4C), all gave the similar results for adults experiencing a constant risk. However, for adults experiencing an exponentially

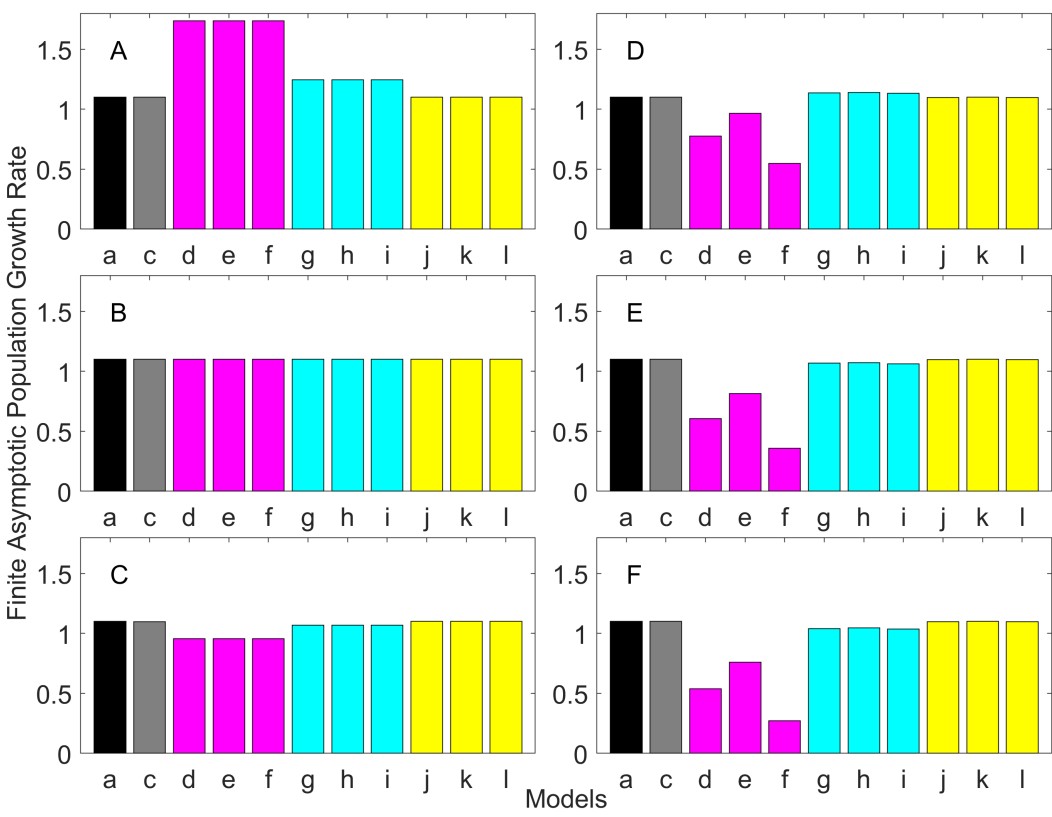

**Figure 4** **Finite asymptotic population growth rate λ when the true λ is 1.100 for early maturation types.** Each panel represents different life history type as was defined in Table 1. Each bar represents a different model for estimating λ. See Table 4 for model types. (Black, Euler-Lotka equation; Gray, age-structured model; Magenta, stage-structured with ordinary mean; Cyan, stage-structured with weighted mean; Yellow, stage-structured with weight mean with discount). (A) Constant Hazard and Increasing Fecundity, (B) Constant Hazard and Constant Fecundity, (C) Constant Hazard Decreasing Fecundity, (D) Increasing Hazard and Increasing Fecundity, (E) Increasing Hazard and Constant Fecundity, (F) Increasing Hazard and Decreasing Fecundity.

increasing risk (Figs. 2D–2F, 3D–3F and 4D 4F, the three means (Models d–f) gave different results. In particular, the harmonic mean (Model f) grossly underestimated λ, as did the geometric mean (Model d) although to a lesser extent. Among the three, the arithmetic mean (Model e) performed better, but it still underestimated λ.

Weighted means (Models g–i) improved the estimation of λ, compared with unweighted means (Models d–e). For example, when the weighted arithmetic mean was used, declining populations (λ = 0.9) were always identified as declining (λ < 1), and increasing populations (λ = 1.1) were always identified as increasing (λ > 1). The use of discounted weight (Models j–k) further improved the estimation of λ, especially, when λ was not at 1.0 (Figs. 3 and 4).

For the delayed maturation types, two age-structured models and nine stage-structured models were used (Figs. 5–7). The two age-structured models differed in whether the age of maturation was assumed to be fixed at the mean age of maturity (Model b; Eq. (15)) or

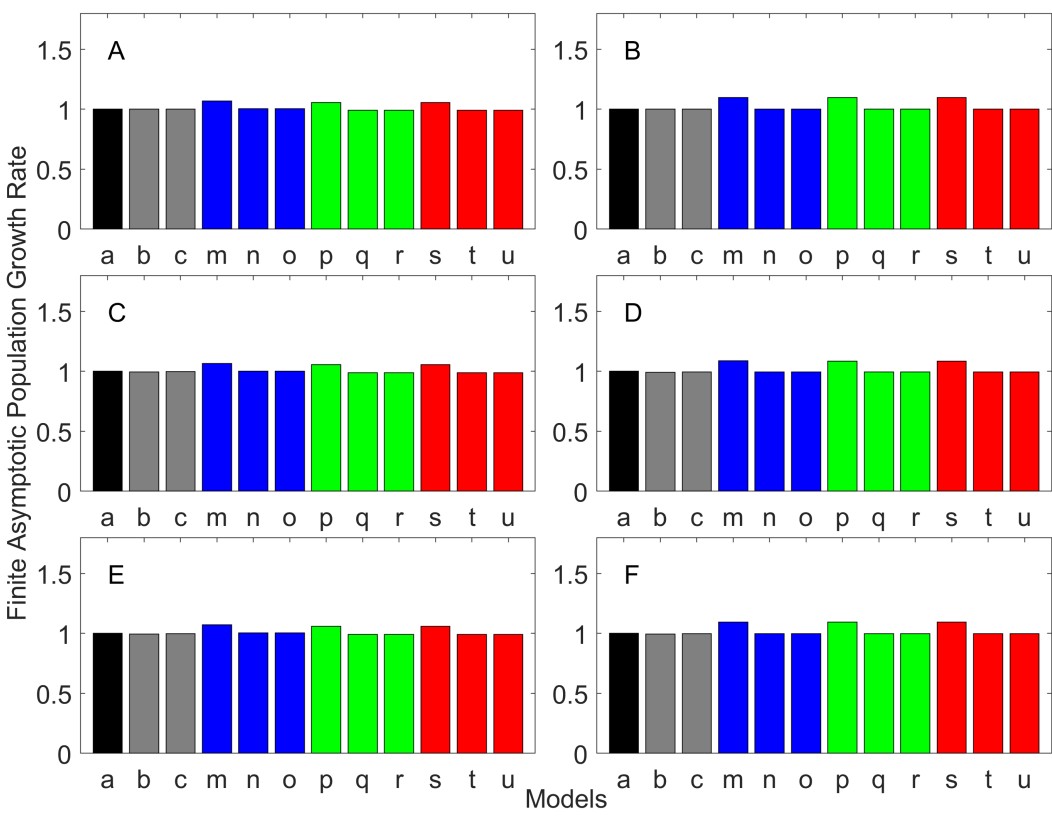

**Figure 5** **Finite asymptotic population growth rate λ when the true λ is 1.000 for delayed maturation types.** Each panel represents different life history type as was defined in Table 1. Each bar represents a different model for estimating λ. See Table 4 for model types. (Black, Euler-Lotka equation; Gray, age-structured; Blue, stage-structured with an arithmetic mean; Green, stage-structured with a weighted arithmetic mean; Red, stage-structured with a discounted-weight arithmetic mean). (A) Decreasing Hazard and Fixed Age of Maturation, (B) Constant Hazard and Fixed Age of Maturation, (C) Decreasing Hazard and Constant Rate of Maturation, (D) Constant Hazard and Constant Rate of Maturation, (E) Decreasing hazard and Increasing Rate of maturation, (F) Constant Hazard and Increasing Rate of Maturation.

the proportion of mature individuals at a given age is incorporated (Model c; Eq. (16)). The nine stage-structured models differed in the method for calculating the conditional transition rate (Table 3) and in how juvenile survival was aggregated (Table 2). For the latter aggregation, arithmetic mean, weighted arithmetic mean, and discounted-weight arithmetic mean were used. First, age-structured models gave λ consistent with the true value (cf. Model a and Models b–c of all panels of Figs. 5–7). This suggests discretization of age did not introduce bias in λ. Furthermore, the incorporation of the proportion of mature individuals did not improve λ (cf. Model b and c of all panels of Figs. 5–7), suggesting the assumption of the fixed age of maturation was appropriate.

For stage-structured models, bias introduced in the estimation of λ was not as large as that with the early maturation types. Among the three conditional transition rate estimation methods (Table 3), the third method (T3) to match the proportion of individuals making the transition after discounting with λ (Models o, r, and u) performed best, and the first

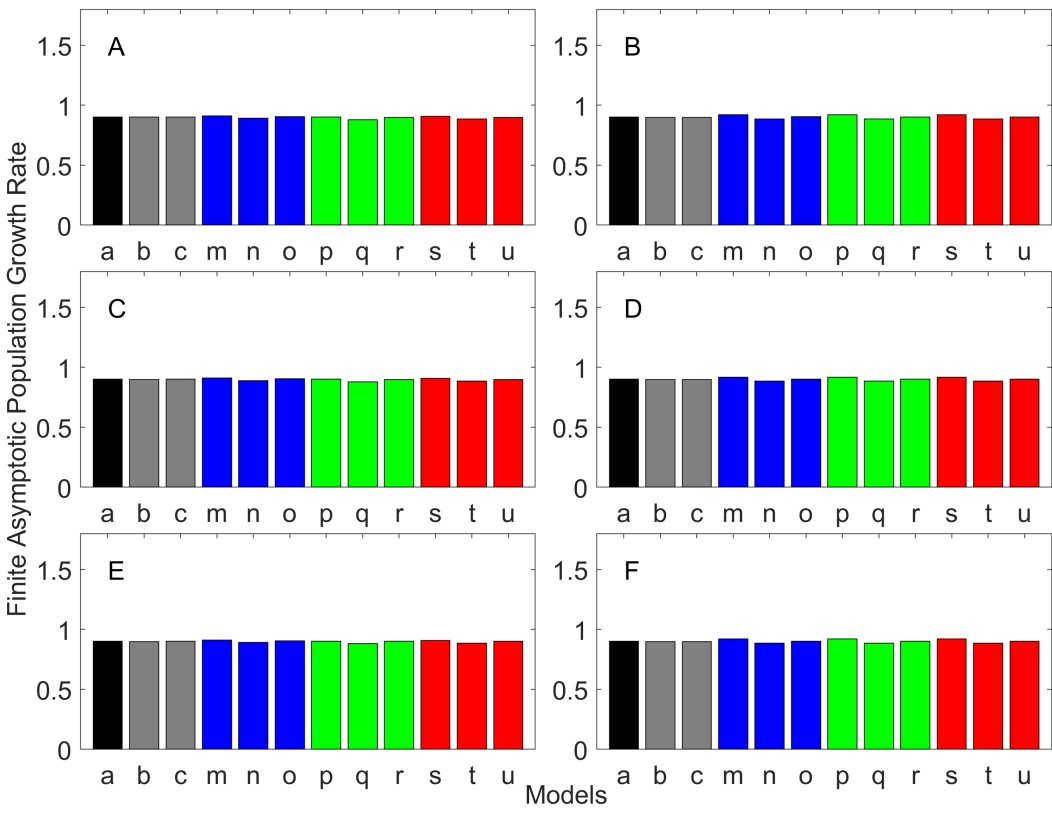

**Figure 6** **Finite asymptotic population growth rate λ when the true λ is 0.900 for delayed maturation types.** Each panel represents different life history type as was defined in Table 1. Each bar represents a different model for estimating λ. See Table 4 for model types. (Black, Euler-Lotka equation; Gray, age-structured; Blue, stage-structured with an arithmetic mean; Green, stage-structured with a weighted arithmetic mean; Red, stage-structured with a discounted-weight arithmetic mean). (A) Decreasing Hazard and Fixed Age of Maturation, (B) Constant Hazard and Fixed Age of Maturation, (C) Decreasing Hazard and Constant Rate of Maturation, (D) Constant Hazard and Constant Rate of Maturation, (E) Decreasing hazard and Increasing Rate of maturation, (F) Constant Hazard and Increasing Rate of Maturation.

method to match the average duration (Models m, p, and s) performed the worst. The estimation was less sensitive to how the juvenile survival rate was aggregated.

## Generation time

Generation time is shown in Figs. 8–13. In all cases, age-structured models (Model b in Figs. 8–10 and Models b and c in Figs. 11–13) gave a generation time that was consistent with the true values (Model a). This suggests discretization of age does not introduce bias. However, stage-structured models gave substantially different values for most scenarios. They gave similar results to the true values only for populations experiencing constant hazard and constant fecundity for early maturation types as long as the population was steady ($\lambda = 1$, Fig. 8B) or growing ($\lambda > 1$, Fig. 10B). These results suggest generation time obtained from stage-structured models are generally not accurate.

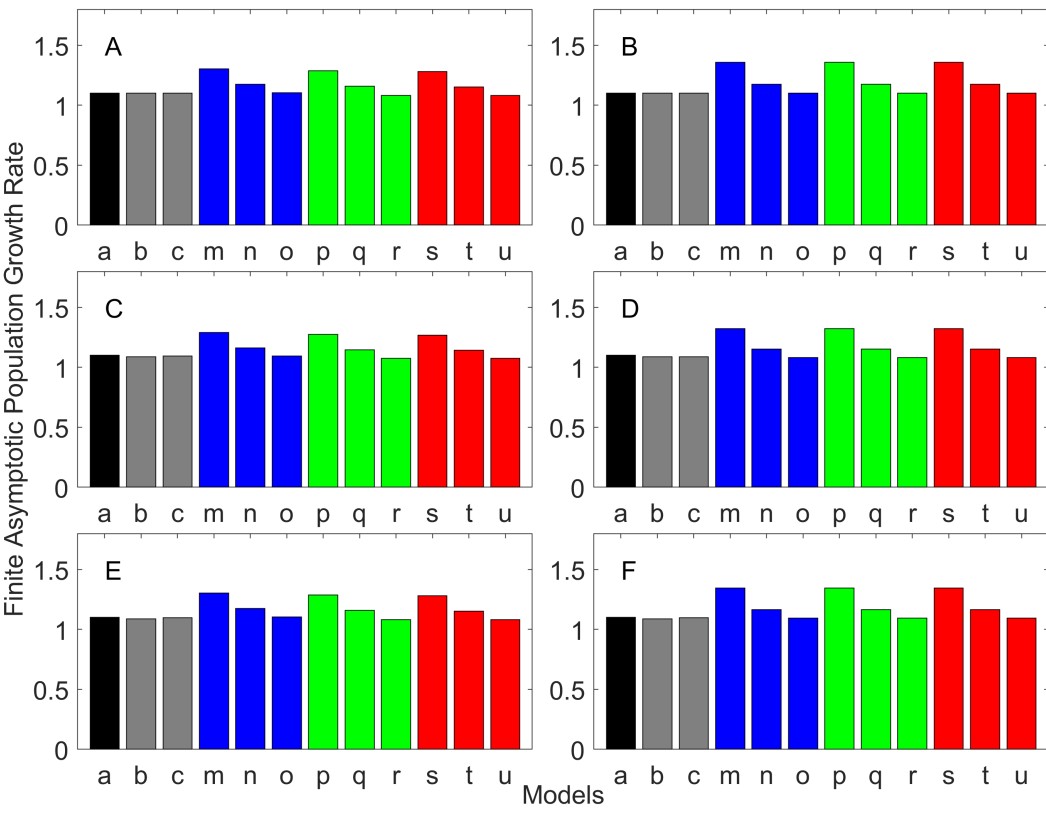

**Figure 7** **Finite asymptotic population growth rate λ when the true λ is 1.100 for delayed maturation types.** Each panel represents different life history type as was defined in Table 1. Each bar represents a different model for estimating λ. See Table 4 for model types. (Black, Euler-Lotka equation; Gray, age-structured; Blue, stage-structured with an arithmetic mean; Green, stage-structured with a weighted arithmetic mean; Red, stage-structured with a discounted-weight arithmetic mean). (A) Decreasing Hazard and Fixed Age of Maturation, (B) Constant Hazard and Fixed Age of Maturation, (C) Decreasing Hazard and Constant Rate of Maturation, (D) Constant Hazard and Constant Rate of Maturation, (E) Decreasing hazard and Increasing Rate of maturation, (F) Constant Hazard and Increasing Rate of Maturation.

## Iterative methods

All of the iterative methods in this study converged very quickly. The estimated λ value obtained using the iterative method is shown in Table 5. Overall, the iterative method performed well in estimating λ. It was accurate to the first digit of λ at least. However, generation time was estimated almost equally poorly with that obtained assuming λ was known *a priori* (i.e., Model u of all panels of Figs. 11–13).

## DISCUSSION

Population growth rate (λ) and generation time are two of the most important pieces of information in determining the status of threatened species (see *IUCN, 2012*). The former tells us how quickly a population is expected to be declining or growing over time, and the latter tells us an appropriate time-scale for a population. Accurate determination of these parameters is essential. For example, the overestimation of generation time or

**Table 4  Model types for estimating $\lambda$ and generation time.**

| Models | Description | Generation time calculation |
|---|---|---|
| a | Euler-Lotka equation | $G_1$ or $G_2$ |
| b | Age structured model with proportion mature | $G_3$ |
| c | Age structured model with fixed age of maturity | $G_3$ |
| d | Two-stage model with a geometric mean survival rate | $G_3$ |
| e | Two-stage model with an arithmetic mean survival rate | $G_3$ |
| f | Two-stage model with a harmonic mean survival rate | $G_3$ |
| g | Two-stage model with a weighted geometric mean survival rate | $G_3$ |
| h | Two-stage model with a weighted geometric mean survival rate | $G_3$ |
| i | Two-stage model with a weighted harmonic mean survival rate | $G_3$ |
| j | Two-stage model with a weighted geometric mean survival rate with $\lambda$ used as discounting factor | $G_3$ |
| k | Two-stage model with a weighted geometric mean survival rate with $\lambda$ used as discounting factor | $G_3$ |
| l | Two-stage model with a weighted harmonic mean survival rate with $\lambda$ used as discounting factor | $G_3$ |
| m | Three-stage model with an arithmetic mean survival rate and matching duration (T1) | $G_3$ |
| n | Three-stage model with an arithmetic mean survival rate and matching proportion transitioning (T2) | $G_3$ |
| o | Three-stage model with an arithmetic mean survival rate and matching proportion transitioning with discount (T3) | $G_3$ |
| p | Three-stage model with a weighted arithmetic mean survival rate and matching duration (T1) | $G_3$ |
| q | Three-stage model with a weighted arithmetic mean survival rate and matching proportion transitioning (T2) | $G_3$ |
| r | Three-stage model with a weighted arithmetic mean survival rate and matching proportion transitioning with discount (T3) | $G_3$ |
| s | Three-stage model with a discounted-weight arithmetic mean survival rate and matching duration (T1) | $G_3$ |
| t | Three-stage model with a discount-weight arithmetic mean survival rate and matching proportion transitioning (T2) | $G_3$ |
| u | Three-stage model with a discount-weight arithmetic mean survival rate and matching proportion transitioning with discount (T3) | $G_3$ |

**Table 5  Population growth rate estimated using the iterative method.** Six different life history strategies of organisms with delayed maturation at three different levels of true $\lambda$ were investigated.

| True $\lambda$ | Life history types (see Table 1) | | | | | |
|---|---|---|---|---|---|---|
|  | D.H-F.M. | C.H.-F.M. | D.H.-C.M. | C.H.-C.M. | D.H.-I.M. | C.H.-I.M. |
| $\lambda = 0.90$ | 0.900 | 0.903 | 0.899 | 0.902 | 0.900 | 0.902 |
| $\lambda = 1.00$ | 0.994 | 1.000 | 0.992 | 0.995 | 0.994 | 0.999 |
| $\lambda = 1.10$ | 1.090 | 1.100 | 1.085 | 1.089 | 1.090 | 1.096 |

underestimation of $\lambda$ will result in species being erroneously placed in categories of higher threat than they should be. In this study, we have evaluated the performance of stage-structured population models in calculating $\lambda$ and generation time, when life table data are used to parameterize the models.
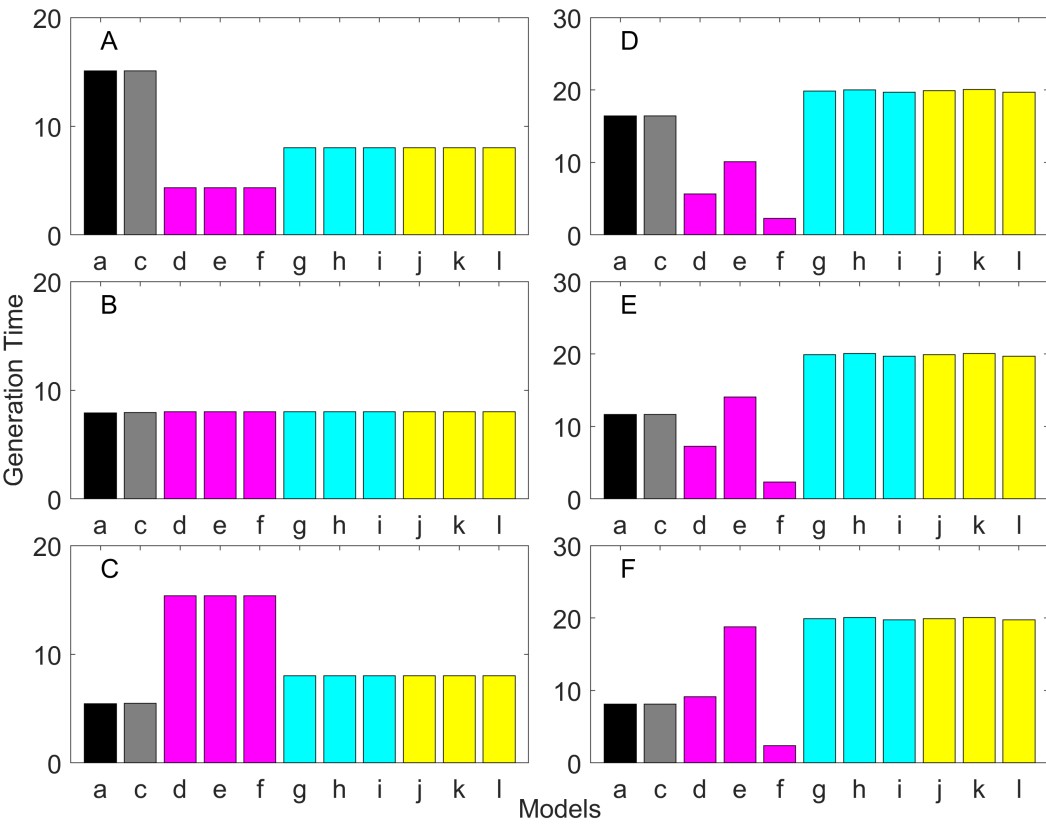

**Figure 8  Generation time for the early maturation types where the true finite population growth rate λ is 1.000 for delayed maturation types.** Each panel represents different life history type as was defined in Table 1. Each bar represents a different model for estimating λ and generation time. See Table 4 for model types. (Black, Euler-Lotka equation; Gray, age-structured model; Magenta, stage-structured with ordinary mean; Cyan, stage-structured with weighted mean; Yellow, stage-structured with weight mean with discount). (A) Constant Hazard and Increasing Fecundity, (B) Constant Hazard and Constant Fecundity, (C) Constant Hazard Decreasing Fecundity, (D) Increasing Hazard and Increasing Fecundity, (E) Increasing Hazard and Constant Fecundity, (F) Increasing Hazard and Decreasing Fecundity.

Overall, discretization of age does not introduce much bias in population growth rate (λ) or generation time (comparing the first and second bars of all panels of Figs. 2–13). Similarly, assuming a fixed age of maturation at the mean age of maturity, does not introduce much bias (comparing the second and third bars of all panels of Figs. 5–7 and 11–13). Although only three types of maturation schedule have been investigated in this study, the latter conclusion is expected to remain true as long as the age-range of maturation is not substantially wide. On the other hand, the aggregations of age-specific vital rates into a stage-structured vital rate introduces bias.

In order to aggregate survival rates for stage-structured population matrices, the use of the discounted-weight arithmetic mean is the most robust method to estimate λ (Figs. 2–4). However, the method requires the use of λ to discount the weight, which defeats the purpose of constructing a stage-structured population model to obtain λ. Therefore, the use of the weighted arithmetic mean is a better option although it can still introduce

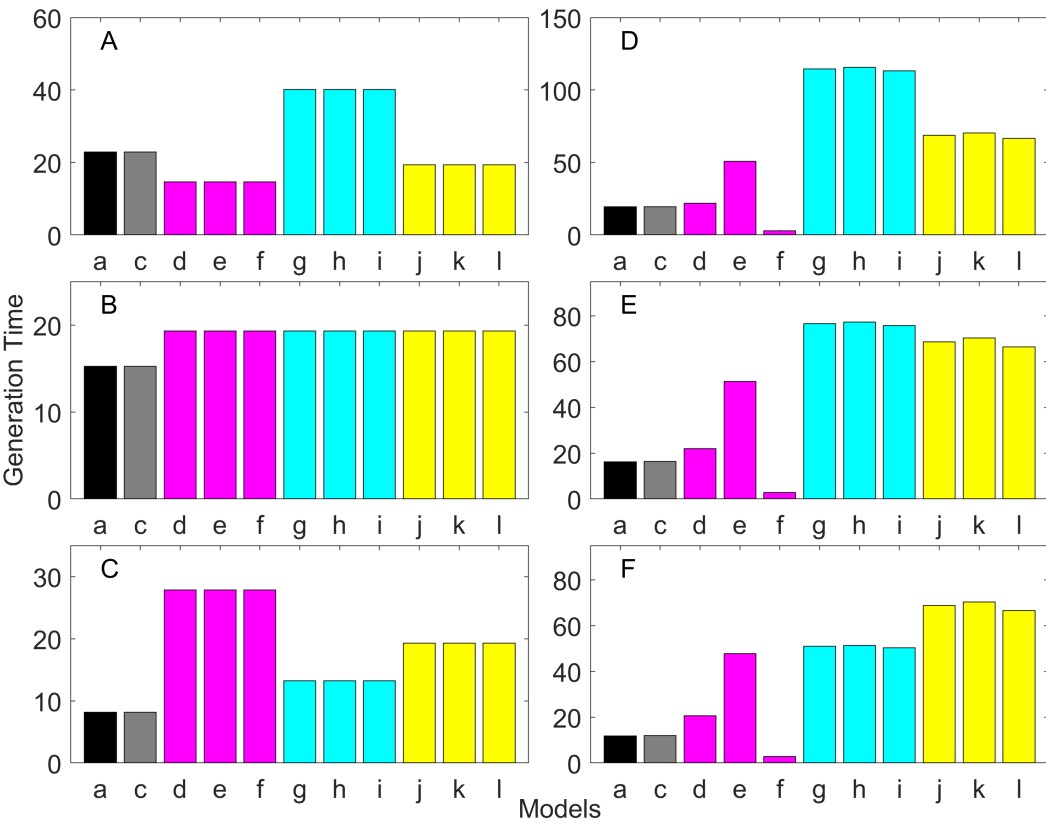

**Figure 9 Generation time for the early maturation types where the true finite population growth rate λ is 0.900.** Each panel represents different life history type as was defined in Table 1. Each bar represents a different model for estimating λ and generation time. See Table 4 for model types. (Black, Euler-Lotka equation; Gray, age-structured model; Magenta, stage-structured with ordinary mean; Cyan, stage-structured with weighted mean; Yellow, stage-structured with weight mean with discount). (A) Constant Hazard and Increasing Fecundity, (B) Constant Hazard and Constant Fecundity, (C) Constant Hazard Decreasing Fecundity, (D) Increasing Hazard and Increasing Fecundity, (E) Increasing Hazard and Constant Fecundity, (F) Increasing Hazard and Decreasing Fecundity.

some bias when the population is either growing or declining. For the calculation of the conditional transition rate, the best approach is to match the proportion transitioning, with λ being used as a discounting factor; however, the use of the method without the discounting factor also performs well (Figs. 5–7).

The study also suggests that aggregating adult stage needs to be done more carefully than aggregating juvenile stage (compare Figs. 2–4 and 5–7). Adults experience survival and reproduction whereas juveniles experience survival and maturation. Because they experience different population processes, it is not surprising to see the difference. However, the results may also be because the adult stage is much longer than the juvenile stage in this study. For example, some organisms exhibit a short-lived adult stage with delayed maturation. For such organisms, the aggregation for the juvenile stage would need to be done more carefully than for the adult stage.

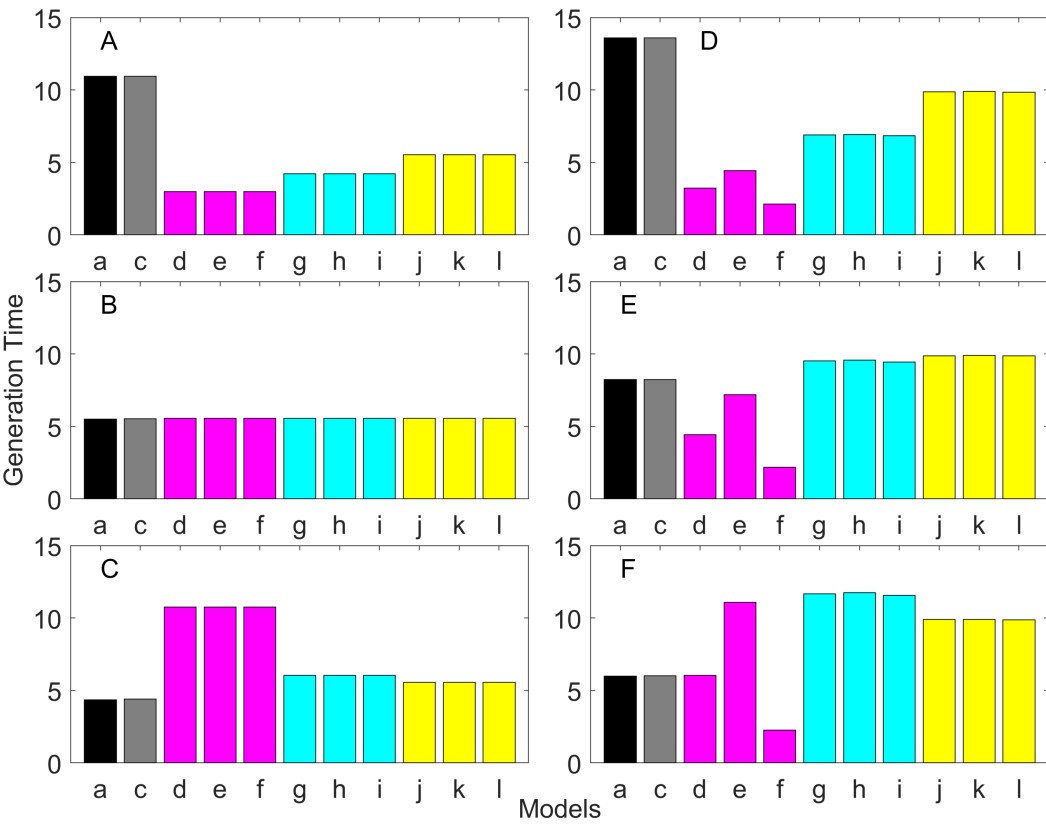

**Figure 10 Generation time for the early maturation types where the true finite population growth rate λ is 1.100.** Each panel represents different life history type as was defined in Table 1. Each bar represents a different model for estimating λ and generation time. See Table 4 for model types. (Black, Euler-Lotka equation; Gray, age-structured model; Magenta, stage-structured with ordinary mean; Cyan, stage-structured with weighted mean; Yellow, stage-structured with weight mean with discount). (A) Constant Hazard and Increasing Fecundity, (B) Constant Hazard and Constant Fecundity, (C) Constant Hazard Decreasing Fecundity, (D) Increasing Hazard and Increasing Fecundity, (E) Increasing Hazard and Constant Fecundity, (F) Increasing Hazard and Decreasing Fecundity.

One of the problems is the heterogeneity among individuals within a stage. In our model, the heterogeneity exists because either the risk of mortality or the fecundity rate changes with age. However, a matrix population model assumes that all individuals are the same within a stage. When both mortality risk and fecundity rate are constant over age, λ and generation time are estimated accurately with stage-structured population models (see second columns of Figs. 2 and 7). However, variation in mortality with age is commonly experienced. For example, young individuals may grow out of a predation risk, and older individuals may experience senescence. The former is common among any animal populations, and the latter is common among long-lived mammals. Fecundity also changes with age. For example, it is often a function of the size of adults, which increases with age, a common change seen in fish. One way to overcome the issue of a non-constant mortality risk or non-constant fecundity may be to aggregate smaller number of age classes into one stage by including a large number of stages in a model. This could be done after

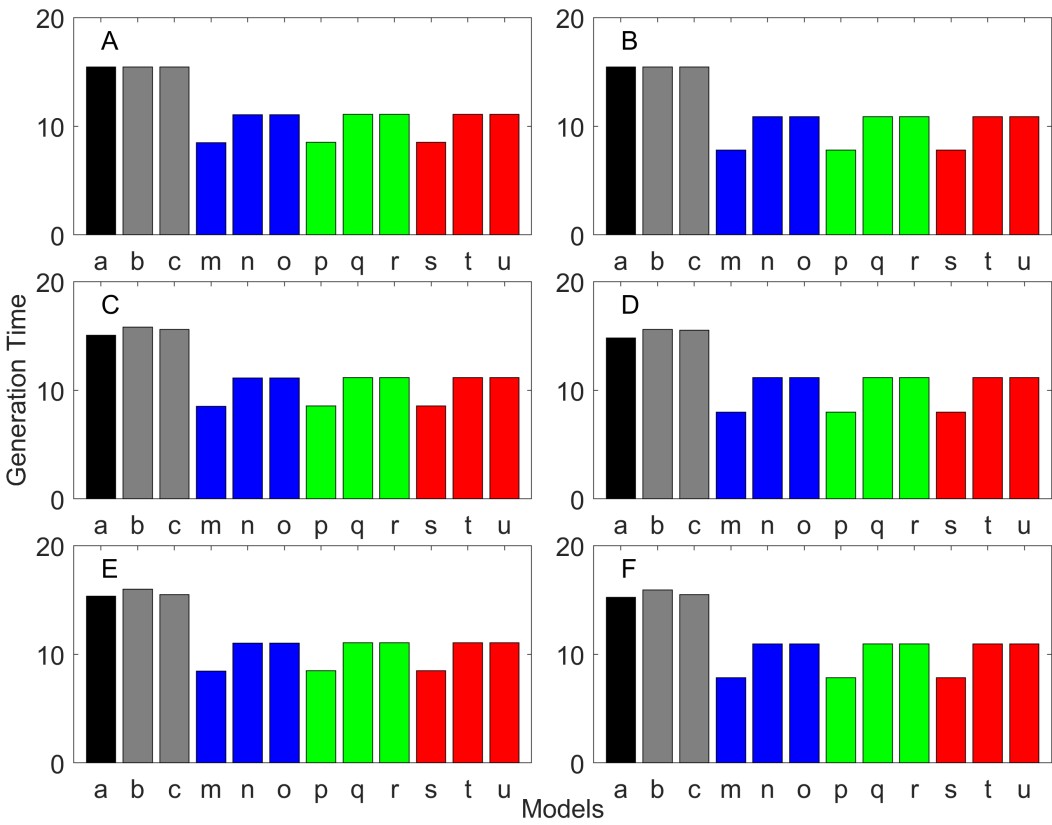

**Figure 11  Generation time for the delayed maturation types where the true finite population growth rate λ is 1.000.** Each panel represents different life history type as was defined in Table 1. Each bar represents a different model for estimating λ and generation time. See Table 4 for model types. (Black, Euler-Lotka equation; Gray, age-structured; Blue, stage-structured with an arithmetic mean; Green, stage-structured with a weighted arithmetic mean; Red, stage-structured with a discounted-weight arithmetic mean). (A) Decreasing Hazard and Fixed Age of Maturation, (B) Constant Hazard and Fixed Age of Maturation, (C) Decreasing Hazard and Constant Rate of Maturation, (D) Constant Hazard and Constant Rate of Maturation, (E) Decreasing hazard and Increasing Rate of maturation, (F) Constant Hazard and Increasing Rate of Maturation.

fitting a competing risk model (*Siler, 1979*) and a flexible fecundity function to life table data and examining the age classes with a similar mortality risk and fecundity.

Our results suggest estimating generation time is more problematic than estimating λ. For example, when fecundity was declining with age under the early maturation model (Fig. 7C), generation time was always overestimated with stage-structured models. This is because although in reality younger individuals contribute more to reproduction than older individuals do, stage-structured models assume homogeneity among adults. In other words, older adults contribute more in the models than the true population. Consequently, generation time is overestimated. When fecundity was declining with age (Fig. 7A), the opposite effect was observed. When survival rate was changing with age, it can lead to over- or under-estimation of generation time, and it is difficult to predict. For these reasons,

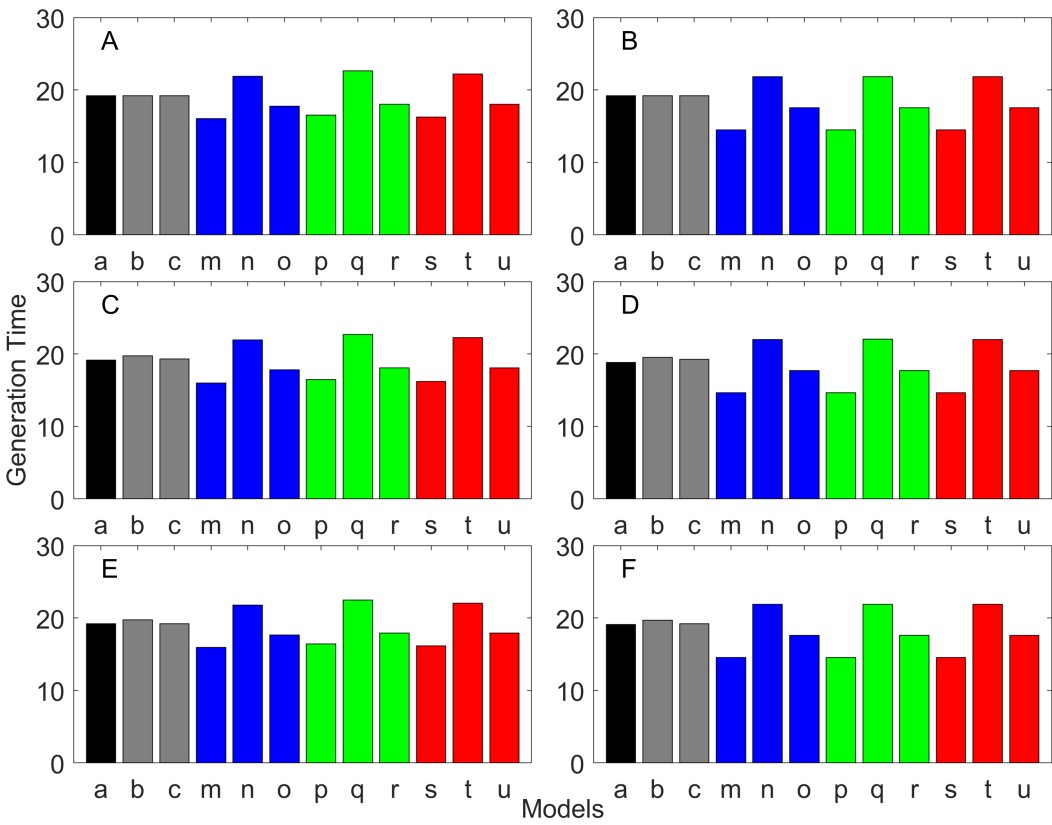

**Figure 12 Generation time for the delayed maturation types where the true finite population growth rate λ is 0.900.** Each panel represents different life history type as was defined in Table 1. Each bar represents a different model for estimating λ and generation time. See Table 4 for model types. (Black, Euler-Lotka equation; Gray, age-structured; Blue, stage-structured with an arithmetic mean; Green, stage-structured with a weighted arithmetic mean; Red, stage-structured with a discounted-weight arithmetic mean). (A) Decreasing Hazard and Fixed Age of Maturation, (B) Constant Hazard and Fixed Age of Maturation, (C) Decreasing Hazard and Constant Rate of Maturation, (D) Constant Hazard and Constant Rate of Maturation, (E) Decreasing hazard and Increasing Rate of maturation, (F) Constant Hazard and Increasing Rate of Maturation.

the construction of an age structured model or the use of the Euler-Lotka equation is recommended for calculating λ and generation time.

*Lebreton (2005)* discusses potential problems with using stage-structured population models to calculate generation time. Lebreton recommends the use of stage-structured models in which stages are embedded within age classes. This effectively increases the number of stages in a model. Such a population matrix can be constructed if one collects information on the stage of individuals in addition to age in order to construct a life table in which individuals are categorized into different age and stage classes. This approach is also useful when life history strategies are complex or when they do not exhibit age-dependent changes in vital rates. For such organisms, a simple life table alone may not be informative because each age class can include multiple stages, but a stage-structured model embedded in age structure should provide accurate estimations of both λ and generation time.
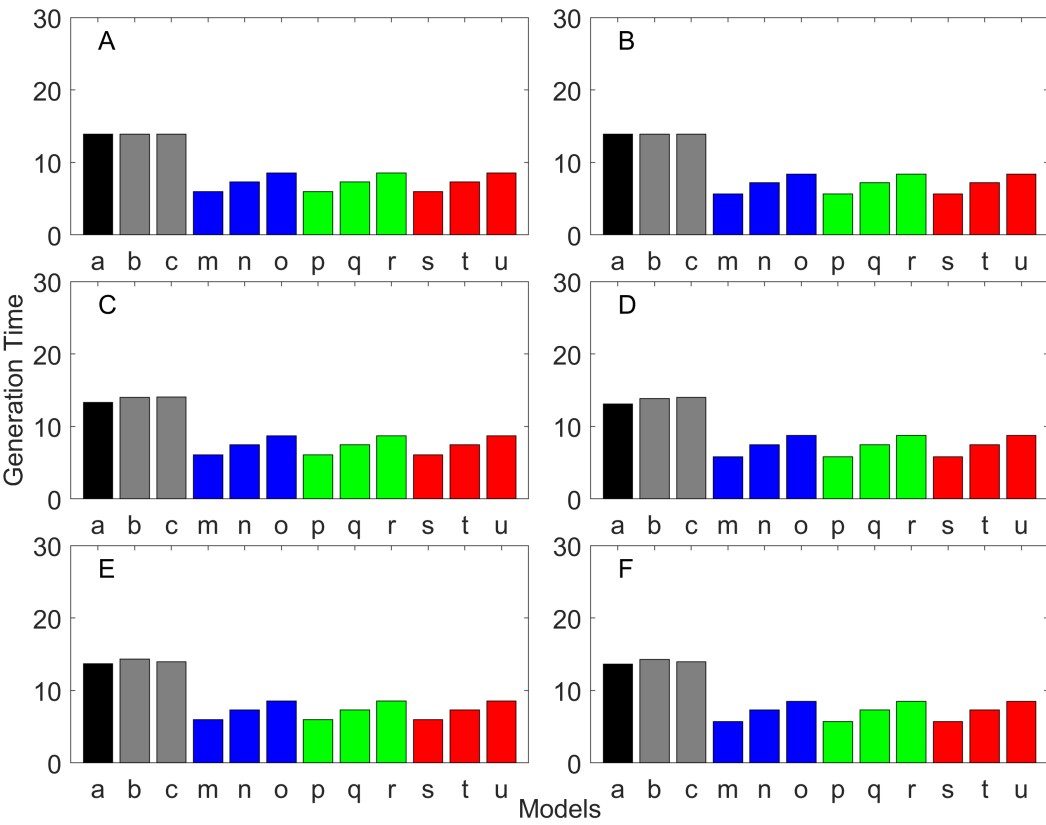

**Figure 13** **Generation time for the delayed maturation types where the true finite population growth rate λ is 1.100.** Each panel represents different life history type as was defined in Table 1. Each bar represents a different model for estimating λ and generation time. See Table 4 for model types. (Black, Euler-Lotka equation; Gray, age-structured; Blue, stage-structured with an arithmetic mean; Green, stage-structured with a weighted arithmetic mean; Red, stage-structured with a discounted-weight arithmetic mean). (A) Decreasing Hazard and Fixed Age of Maturation, (B) Constant Hazard and Fixed Age of Maturation, (C) Decreasing Hazard and Constant Rate of Maturation, (D) Constant Hazard and Constant Rate of Maturation, (E) Decreasing hazard and Increasing Rate of maturation, (F) Constant Hazard and Increasing Rate of Maturation.

*Salguero-Gomez & Plotkin (2010)* also discusses the relationship between the number of stages and various statistics obtained from stage-structured population matrices.

An additional problem is the change in stage/age distribution due to growing or declining population abundance. A population growth rate needs to be discounted when a survivorship curve is estimated from a life table. Discounting is required for the same reason that it is included in the Euler-Lotka equation. There are two types of life table, dynamic or static life table. A dynamic life table is also called a cohort life table because it is constructed using data obtained by following the same cohort over time/age. On the other hand, a static life table is obtained by examining the age distribution of samples collected at one sampling occasion. Throughout this paper, a dynamic life table has been assumed. When a static life table is collected instead, a survivorship curve needs to be discounted by a population growth rate to obtain the true survivorship curve. The discounting is necessary

because individuals in different age classes in the data were born at different time and the population is growing or declining so that increasing or decreasing numbers of individuals are born from one year to the next. Interestingly, the age distribution obtained directly from a static life table gives the discounted weight used in this study.

The iterative method allows the use of $\lambda$ in constructing a stage-structured population matrix without knowing $\lambda$ *a priori*. This method performed well in identifying whether $\lambda$ is greater or smaller than 1. In this sense, the iterative method is a viable option for constructing a stage-structured population model. However, when a life table is available, an age-structured matrix population model can be used for calculating $\lambda$ more accurately. Furthermore, an age-structured matrix allows accurate estimation of generation time, which can be substantially biased in stage-structured population models.

In this study, bias was introduced when constructing matrix population models using information from life table data. As an alternative to life table data, Cormack–Jolly–Seber (CJS) type capture-recapture data (*Lebreton et al., 1992*) can be obtained. With additional information to categorize captured individuals into different stage classes, a multi-type/multi-stage capture recapture method can be used for estimating parameters in stage-structured matrix population models directly (*Fujiwara & Caswell, 2002*; *Nichols et al., 1992*). Provided there is no heterogeneity in a capture rate within a given stage, the method accounts for an underlying age-distribution although an actual age distribution may not be observable. When age is not known, the use of stage-structured population models is the only option. However, generation time obtained from such models may not be accurate. Future investigations of the performance of generation time calculations with stage-structured population models when vital rates are estimated from individual capture-recapture data are needed.

## CONCLUSIONS

When life table data are collected, we recommend fitting a competing risk model and a flexible fecundity function to the data and estimating population growth rate $\lambda$ and generation time using an age-structured population matrix or the Euler-Lotka equation. Calculating generation time using stage-structured population models should be avoided. If a researcher is interested in constructing stage-structured population models (e.g., for the purpose of sensitivity and elasticity analyses), the conversion from age-structured vital rates to stage-structured vital rates should be done by aggregating age classes with similar mortality risk and fecundity into the same stage. When aggregating survival rates for constructing stage-structured population models, discounted-weight arithmetic mean should be used. When calculating the conditional transition rate, one should use the method for matching the proportion making transitions with $\lambda$ as a discounting factor.

### Funding
Jasmin Diaz-Lopez was funded by the Applied Biodiversity Science Conservation Scholars Program through the support of the National Fish and Wildlife Foundation. There was no additional external funding received for this study.

### Grant Disclosures
The following grant information was disclosed by the authors:
Applied Biodiversity Science Conservation Scholars Program.
National Fish and Wildlife Foundation.

### Competing Interests
The authors declare there are no competing interests.

### Author Contributions
- Masami Fujiwara conceived and designed the experiments, performed the experiments, analyzed the data, contributed reagents/materials/analysis tools, wrote the paper, prepared figures and/or tables, reviewed drafts of the paper.
- Jasmin Diaz-Lopez conceived and designed the experiments, analyzed the data, wrote the paper, reviewed drafts of the paper.

### Data Availability
The complete code used for generating all of the results (along with parameters) has been provided as Supplemental Files.

### Supplemental Information
Supplemental information for this article can be found online at http://dx.doi.org/10.7717/peerj.3971#supplemental-information.

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
