# Peer review of "Constructing stage-structured matrix population models from life tables: comparison of methods"

_PeerJ, doi:10.7717/peerj.3971_

## Round 0.1 · original submission · Minor Revisions

· Academic Editor

Minor Revisions

You have reports from three expert reviewers. All of them make constructive comments.

Please address these comments in your response memo if you choose to resubmit (which I assume you will). When I say "address" I do not mean you must adhere to each and every comment to gain acceptance, but in cases where you do not agree, you must provide a rationale. The comments strike me as sensible so you have a great opportunity to improve the paper.

Please also change the personal pronoun to "we" from "I", since this is a multi-author paper. (At least one of the referees also noted this.)

Reviewer 1 ·

Basic reporting

no comment

Experimental design

no comment

Validity of the findings

no comment

Additional comments

Review of ``Constructing stage-structured matrix population models from
life tables: Comparison of method" by Masami Fujiwara and Jasmin Diaz-Lopez

Stage based demographic models are useful when life stages other than age of an individual
better capture survival and reproductive rates. Such models have been discussed in
ecological and demographic literature for some time though issues related to construction and
interpretation have persisted. This ms is a useful contribution that addresses a major issue
in linking age-based models with stage-based models. It provides some general clues about when stage-based models would work and provide guidelines for construction of such models.

Here are a few suggestions that I hope would help in improving the readability of the ms:

1) I would like to see a bit more clarity in the Discussion section about the discrepancy between age- and stage-based models with regard to estimates of $\lambda$ and generation time. It is shown that generation time as opposed to $\lambda$, is largely way off when derived from stage-based models.
Can authors provide some intuitive reasoning into why this is so? Is this a numerical issue or is there anything problematic when using equations 13-14 for stage-based models? Since only in the case of constant hazards (and constant fertility) stage-base models do well I am wondering which, the aggregation of age-groups to make stages, or individual heterogeneity contribute more to the error in estimates. Can this information be dished out from the results?

2) The ms is generally well written though it was hard for me to go back and forth between the text and the 16 tables at the end of the ms.
I would suggest converting the tables into figures if possible. You could plot deviations between estimates and true values, for instance. I would also think of moving some of the model details (equations 1-20) to an appendix leaving only qualitative information in the text. Please give citations to the different survival/mortality and fertility functions if they have been used previously. I would also suggest providing the codes as some of the methods used to compare different approaches can be used as tools to make models when data is available.

3) Line 177: why is $l(x) = n_1(x)+ n_2(x)$? If so, is it guaranteed that $l(x) \le 1$ for all $x$, as $n_1$ and $n_2$ are population densities?

4) Line 232: should be $l(z)l^{-1}(x-0.5)$?

5) Line 237: what is $l(0.5)$?

6) Line 269 (and 278): what are the symbols L, M and O etc. inside the matrix? Never explained, I think.

7) Line 384: I am not sure if anything is said about the convergence; may be good to provide some details about how fast the methods converged etc.

8) Line 395: check sentence: ``higher threat than"

9) Line 403: ``age-range"

10) Line 453: check sentence

11) Lines 461-467: this paragraph is not very informative and seems redundant.

12) Line 481: ``We" recommend..(unless both the authors listed are same!)

13) The title should be "Comparison of methods" I guess (not "method")

Reviewer 2 ·

Basic reporting

The article is mostly clear, properly referenced, and put in adequate context.

Presenting the main results in a long series of tables is not the most effective strategy for communicating the results. I think you could use histograms to summarize results for lambda and generation time, perhaps also split by life-history type. Color coding could be used to embed more information into such histograms, e.g differentiate different types of averaging, or whatever you deem worth highlighting.

The language is a bit imprecise in some places. This issue is most serious for the terminology related to rates, ratios, and probabilities. To me a "rate" is something that has dimension 1/time. This manuscript starts with introducing "rates" that are not "true rates" thus defined. In a discrete time model, survival is not a rate but a probability. Reproduction and lambda are ratios. Of course, I readily admit that the use of term "rate" is often quite liberal, and people have tendency to include even when it is not needed or is "wrong" (I have also done so in the past). This is not necessarily a big issue - when the context is clear, they may not be a risk of confusion. But in this article some "rates" are "true rates" whereas others are ratios or probabilities. The text would greatly benefit if you could clean up this terminology.

Experimental design

This manuscript presents a useful comparison of different ways of aggregating data for stage-structured population models. The design of the analyses is adequate for the task.

Validity of the findings

No comments

Additional comments

Specific comments:
L19, abstract: I would shorten this, even though it probably is within the allowed limits. You would want to provide a concise overview of your work
L33: discretization ... introduce*s*
L44: "life history types" (simpler and less clumsy)
L47: "I recommend" should be "we recommend", assuming that both authors stand behind this recommendation.
L74: remove "However" that hints to a non-existent contrast
L90: usually simply "Euler-Lotka equation". In any case, no caps for "Renewal Equation" (also later)
L94 and below: why "a renewal equation" when you specifically have defined it as referring to the Euler-Lotka equation?
L105: remove the decimal from "1.0" because the value is exactly "1"
L110: here you should write "*the* renewal equation"
L123: maybe you should define what "the second stage" is?
L132: I would call this "mortality rate" (this is a true rate because its unit is reciprocal of [x])
L134: why alpha_3 and beta_3 here, and alpha_2 and beta_2 in eq. 6 introduced later? If the notation has no mnemonic basis, at least the ordering should be logical.
L141: this equation is missing a scaling parameter (fecundity per length^3), otherwise R in this equation is completely different from R in equations 4 and 5. I would subscript the R's differently anyway because the exact interpretation is different even after standardizing the units.
L206: as L105
L269,278: in my pdf, I see L, M and O where there probably should be ellipses
L343: Should this be "similar"? They are the same under the chosen precision. Either say that or use "similar". Also further below - "same" literally means identical and should not be used loosely.
L380: you need to add "with the true values", otherwise "consistent" is ambiguous (or better "similar to the true values ")
L387: I don't understand what "and they performed poorly" refers to
L390: pompous, "two key parameters describing population demography" or something would be better
L390: insert "expected to be" before "declining"
L392: of population, not species
L414: there is a fundamental difference in that adults reproduce and survive while juveniles only survive, so the situation is unlikely to be symmetric
L429: only when growth is indeterminate, which is typically true for fish, but also for many other cold-blooded vertebrates as well as many invertebrates

Reviewer 3 ·

Basic reporting

The authors studied the convertibility between age-structured and stage-structured models with a life table and examined the convertibility using two basic demographic statistics. i.e., population growth rate and generation time. The description of the introduction is easy to understand the background of the issue they want to solve. It is written thoroughly citing many related references. The purpose is clear, that is, to investigate the performance of methods for such conversions. The manuscript is organized well according to PeerJ standards. The figures and tables are well-organized and well-labeled such that the readers can understand what they meant.

Experimental design

The original question is clear and well-defined, i.e., how to extent age-structured model can be converted to stage-structured model in obtaining the population growth rate and generation time. Most researchers, who have used population matrix models and the matrix database, would be interested in it and be eager to know the answer. It is definitely within the scope of the journal. Most of the assumptions made in the calculation are good and sufficiently described, but I have several complaints and concerns on a part of the assumptions (see comment 2 and 3).

Validity of the findings

Their findings, namely the conversion performed well in identifying the population growth rate for many life history strategies of organisms and performed poorly in estimating generation time, is astonishing and interesting. The recommendation in Conclusion is rationale and includes meaningful implication. However, the result on the discrepancy of generation time among the methods are not fully explained and compared (see Comment 1).

Additional comments

(1) The most important point to be revised is on the description in the result of generation time (377l. to 382l.). The result showed that the discrepancy is substantial for most scenarios. However, the explanation is very short and insufficient to understand the discrepancy. Many questions arose when I read the result. Which scenario is the worst (or medium)? Does it depend on the maturation timing (early- and delayed-maturation)? Why does the discrepancy arise? I need more explanation about them in Results and Discussion including your speculation.
(2) My second concern is the different assumptions of survivorship between early maturation type and delayed maturation type. The authors assumed two types of survivorship both for juveniles and adults (Eqs. 1 and 2) in early maturation, and assumed Eqs. 1 and 6 in juveniles and Eq. 1 in adults. I do not understand why the authors change the assumption because I think it is not necessary in this study. The difference leads to the difficulty in comparing the results between early- and delayed- maturations. The comparison must be very valuable.
(3) The authors used 2 by 2 matrix in the construction of stage-structured matrix for early-maturation type. Several researchers cautioned that population growth rates strongly depend on matrix size. The variation in matrix dimensions complicates the comparison among the statistics of the matrices (see Salguero-Gomez 2010). Therefore, I expected that the results in tables 4 to 6 are very different from those in tables 7 to 9. Though they seem to be similar and not to affect the difference so much in your manuscript, it’s better to note lightly in Discussion.
(4) This is just a comment. You explained the result of the estimation of population growth rate on lines 349 to 376. I recommend you to make a summary table in order to advise in which case we have to be careful when we used the converted stage-structured matrices. It would be useful for readers to understand the results at a glance.

Miscellaneous points:
(i) On line 134: The parameter alpha_3 and beta_3 should be alpha_2 and beta_2. The subscripts should be numbered in order of their appearance.
(ii) On line 173: “m_1 = 0.5 for x > x_2” There must be a mathematical contradiction here because x1=x2 in the first type of maturation. The equality symbol is necessary?
(iii) On line 197: the stage -→ stage i
(iv) On lines 231 and 233: l^-1(x-0.5) appears the product of l^-1 and (x=0.5). It’s better to use the fractional form here. Is the b(x)in the equation b(z)?
(v) On line 260: Equation 13 is a typical formula of generation time. Please cite a good textbook on demography. Eqs.13, 14 and 18 are denoted as G using the same symbol. It’s better to denote them as G_1, G_2 and G_3. You should specify which generation time was used in the tables.
(vi) On line 266: at age x, respectively. --→ at age x, respectively (in Eq. (7)).
(vii) On line 269: f_2 -→ f_1.5?
(viii) On line 324: priori? Is it “a priori”?

---

## Round 0.2 · Minor Revisions

· Academic Editor

Minor Revisions

Almost there. You have referee reports from two anonymous reviewers. They point out just a few residual changes they would like to see.

Please consider these changes. We will not do endless loops of revisions, I promise.

Thank you.

Reviewer 2 ·

Basic reporting

The manuscript is reasonably well written but the results are still rather heavy to follow. This is because one has to refer to several figures at the same time, the figures are not made as accessible as they could, and the way of referring to their elements in the text is different from the figures itself. See also my "general comments".

I do not like calling everything as "rates", especially probabilities, but I acknowledge that there is no universal standard for this.

Experimental design

No new comments, this is fine

Validity of the findings

No new comments, conclusions reflect the results without over interpretations

Additional comments

Specific comments:
L41 "stage-structured" (hyphen, for consistency)
L52 Split this sentence after "and" --- the statement being made is not related to the first clause.
L79 It is never "necessary", so I would say something like "little is gained by converting..."
L180 I wonder if there is some reasoning behind using different means, or is it just ad hoc choice?
L322 Tell the reader which columns shoes this
L329 "(Models D-F in panels a-c of Figures 5-7" - the use of lower/upper case letters here is the opposite to their usage in the figures. Check this throughout, very confusing!
L339 The figure ref should be moved to the end of the sentence
L359 Figs. 11-13 do not include model B
L365 This should probably read "are generally not accurate", because they can be accurate in special cases
L391 "enable the estimation of"->estimate
L404 "it is possible that in reality" sounds odd and should be deleted, given that there are lots of insects that do exactly this, periodic cicadas being an extreme example.
L407 I would add here "In our model, the heterogeneity ..." because there are many other reasons, more in general
L478 Here you could advise against estimating generation time from stage-structured population models
L492 Remove the hyphen from "Orcinus-orca"

Table 1 None of the parameters are dimensionless. It would be good to make some statement about the measurement units or dimensions.
Figures:
* Each figure results from a 2x3 factorial design, but the panel order does not reflect this. With three rows and two columns, it would be logical and reader-friendly to let "H" with two levels to vary by column, and "F" with three levels by row. It looks like that you should fill the figures column-wise.
* The figures would probably benefit from marking the "a" column differently from the others, perhaps also adding a horizontal reference line
* More in general, indicating certain types of models with coloring or patterning would probably be helpful. Of course, this should not be overdone.
* Given that the figures 5-7 are structurally identical, you could probably merge them into one, with e.g. grouped bars or whiskers to show results for the three levels of lambda. The same applies to the other figure triplets.

Reviewer 3 ·

Basic reporting

no additional comment.

Experimental design

I suggested two points to be revised (Comments 2 & 3). It's now appropriately revised.

Validity of the findings

I commented "the result on the discrepancy of generation time among the methods are not fully explained and compared". It's now appropriately revised.

Additional comments

Most of my comments are appropriately responded. However, there are still several typos and points to be revised.

Reply to 1st comment: The expansion in discussion is good. It's now appropriately revised, I think.

Reply to 2nd comment: The author probably misunderstood what I meant in the comment. However, clarifying the reason or the mechanism of the difference between early-maturation and delayed-maturation is not a central theme in this paper. I agree with the author’s opinion, i.e. their recommendation is to use age-structured model to calculate lambda and generation time when age-specific vital rates are available.

Reply to 3rd comment: The revised discussion on 399l. to 406l. is enough to answer my comment.

Miscellaneous points:
(i) On line 177: the stage begins at age x_i. I understand “the stage” in the senetnce means stage i in stage-structured model, I think. Otherwise, there is no definition of i.
(ii) In Eqs. (15) and (16): There are many 2s in matrix-element subscript and some of the subscripts are described as 1.5. I do not understand the inconsistency. Which is correct, 2 or 1.5?
(iii) On line 362: They gave similar to the true values -→ They gave similar result to the true values?
(iv) Table 4: You defined three types of generation time, G_1 to G_3. Please describe which generation time is use in which model type.
(v) Figures to 16: Why do you use A to F in the panels. The capital A to U are used to identify the models you used in Table 4. It’s very confusing. The horizontal axes should be described using upper cases?

---

## Round 0.3 · accepted · Accept

· Academic Editor

Accept

You have satisfied the referees and myself that your latest revisions address the previous concerns.

Reviewer 2 ·

Basic reporting

The authors have adequately addressed my residual comments. I have no further comments.

Experimental design

No new comments

Validity of the findings

No new comments

Reviewer 3 ·

Basic reporting

No additional comments.

Experimental design

No additional comments.

Validity of the findings

No additional comments.

Additional comments

No additional comments.